# Deletion and tandem duplications of biosynthetic genes drive the diversity of triterpenoids in *Aralia elata*

Yu Wang [1,2,10], He Zhang [2,3,10], Hyok Chol Ri[2,9,10], Zeyu An[2,10], Xin Wang[2], Jia-Nan Zhou[2], Dongran Zheng[2], Hao Wu[2,4], Pengchao Wang[2], Jianfei Yang[5], Ding-Kun Liu[6], Diyang Zhang [6], Wen-Chieh Tsai [7,8], Zheyong Xue [2,4], Zhichao Xu[2,4], Peng Zhang [5✉], Zhong-Jian Liu [6✉], Hailong Shen [1,5✉] & Yuhua Li [2,3✉]

Araliaceae species produce various classes of triterpene and triterpenoid saponins, such as the oleanane-type triterpenoids in *Aralia* species and dammarane-type saponins in *Panax*, valued for their medicinal properties. The lack of genome sequences of *Panax* relatives has hindered mechanistic insight into the divergence of triterpene saponins in Araliaceae. Here, we report a chromosome-level genome of *Aralia elata* with a total length of 1.05 Gb. The loss of 12 exons in the dammarenediol synthase (DDS)-encoding gene in *A. elata* after divergence from *Panax* might have caused the lack of dammarane-type saponin production, and a complementation assay shows that overexpression of the *PgDDS* gene from *Panax ginseng* in callus of *A. elata* recovers the accumulation of dammarane-type saponins. Tandem duplication events of triterpene biosynthetic genes are common in the *A. elata* genome, especially for *AeCYP72As*, *AeCSLMs*, and *AeUGT73s*, which function as tailoring enzymes of oleanane-type saponins and aralosides. More than 13 aralosides are de novo synthesized in *Saccharomyces cerevisiae* by overexpression of these genes in combination. This study sheds light on the diversity of saponins biosynthetic pathway in Araliaceae and will facilitate heterologous bioproduction of aralosides.

[1] State Key Laboratory of Tree Genetics and Breeding, Northeast Forestry University, Harbin 150040, China. [2] College of Life Sciences, Northeast Forestry University, Harbin 150040, China. [3] Key Laboratory of Saline-alkali Vegetation Ecology Restoration (Northeast Forestry University), Ministry of Education, Harbin 150040, China. [4] Heilongjiang Key Laboratory of Plant Bioactive Substance Biosynthesis and Utilization, Northeast Forestry University, Harbin 150040, China. [5] School of Forestry, Northeast Forestry University, Harbin 150040, China. [6] Key Laboratory of Orchid Conservation and Utilization of National Forestry and Grassland Administration at College of Landscape Architecture, Fujian Agriculture and Forestry University, Fuzhou 350002, China. [7] Orchid Research and Development Center, National Cheng Kung University, Tainan 701 Taiwan, China. [8] Department of Life Sciences, National Cheng Kung University, Tainan 701 Taiwan, China. [9] Present address: Biochemistry Institute, University of Science, Pyongyang 999093, Democratic People's Republic of Korea. [10] These authors contributed equally: Yu Wang, He Zhang, Hyok Chol Ri, Zeyu An. ✉email: zhangpeng@nefu.edu.cn; zjliu@fafu.edu.cn; shenhl-cf@nefu.edu.cn; lyhshen@126.com

*A*ralia species, belonging to Araliaceae, are shrubs or small trees (Supplementary Fig. 1). *Aralia elata*, mainly distributed in Northeast Asia, has been used as a medicinal and edible plant for hundreds of years[1] because it is enriched in active secondary metabolites, including triterpenes, diterpenes, flavonoids, coumarins, phenols, and protopectins[2]. Among them, the triterpenoid saponins, called aralia saponins (aralosides), have been reported to be responsible for the medicinal properties of *A. elata*[3,4]. For example, *Aralia*-specific araloside C, which has been identified only in *Aralia* species, has protective effects on the cardiovascular system; it can attenuate foam-cell formation and lessen atherosclerosis by modulating macrophage polarization via Sirt1-mediated autophagy[5]. Chikusetsusaponin IVa has antiviral activities against herpes simplex virus type 1 (HSV-1), HSV-2, human cytomegalovirus, measles virus, and mumps virus with high selectivity indices[6]. Oleanane-type triterpenoids are biological defensive substances with antibacterial activities against agricultural pests[7,8] and confer antibiotic resistance to some strains[9].

The Araliaceae family consists of approximately 55 genera and more than 1,500 species, most of which are used as oriental medicines and have diverse triterpenoid saponins[10]. Based on morphological and molecular evidence, the Araliaceae family encompasses two major monophyletic groups: the *Asian Palmate* group and the *Aralia-Panax* group[11,12]. The *Aralia-Panax* group consists of two closely related genera, *Aralia* L. and *Panax* L.[13]. Although *Aralia* is closely related to *Panax*, their saponins are very different in type and composition (Fig. 1). In the genus *Panax* L., the famous herbs *Panax ginseng* and *Panax notoginseng* are known to accumulate dammarane-type ginsenosides, such as ginsenosides Rg3 and Rh2, which are widely used as antitumor drugs[14,15]. Notably, dammarane-type triterpenoids have not been isolated in *Aralia*, while over 100 kinds of oleanane-type pentacyclic triterpenes have been recorded, exhibiting an extremely high level of triterpenoid diversity[16]. In contrast, oleanolic saponins are present in low concentrations in *P. ginseng* and *P. notoginseng*[17]. Hence, the structural differences of triterpene saponins in closely related species of *Panax* and *Aralia* are of considerable interest.

All the triterpenoids in plants are initially generated from 2,3-oxidosqualene, which is mainly derived from the cytosolic mevalonic acid (MVA) pathway and probably to a minor extent from the plastidial methylerythritol phosphate (MEP) pathway. Then, 2,3-oxidosqualene is converted into oleanane- and dammarane-type triterpene skeletons by different members of the oxidosqualene cyclases (OSCs), named β-amyrin synthase (BAS) and dammarenediol synthases (DDSs), respectively. The triterpene scaffolds are further oxidized and glycosylated into different structures of triterpenoids by various cytochrome P450-dependent monooxygenases (P450s) and UDP glucosyltransferases (UGTs) or cellulose synthase-like (CSLs)[18], respectively. Recently, the genomes of *P. ginseng* and *P. notoginseng* have been sequenced by several groups[19–24], and the biosynthetic pathways of *Panax* dammarane-type triterpenes have been mostly elucidated[25–27]. However, the key enzymes involved in the synthesis of aralosides in *A. elata* are still unidentified due to the lack of genomic information for this species.

Here, we present a high-quality reference genome of *A. elata* to investigate the evolutionary mechanism of the compositional variations of dammarane-type and oleanane-type triterpenoid saponins between the *Panax* and *Aralia* genera. Using genome-wide identification, gene expression, and biochemical assays, we determine that the functional loss of OSCs and tandem duplication events of CSLs, P450s, and UGTs are crucial for the evolution of the biosynthesis and accumulation of diverse aralosides in *A. elata*. In addition, we introduce the biosynthetic genes into yeast strains to produce various aralosides. This work provides insights into the genomic evolution and speciation of Araliaceae species and provides a valuable foundation for clarifying the evolutionary mechanism of triterpenoid biosynthesis and identifying unknown gene modules for araloside biosynthesis via synthetic biology approaches.

## Results and discussion

**Genome assembly and annotation**. Based on the *K*-mer distribution analysis, we estimated the genome size of diploid *A. elata* to be 1.08 Gb, with a high level of heterozygosity (1.62%) and repetitive sequences (78.38%) (Supplementary Fig. 2). In total, 97.88 Gb of long reads with an average read length of 12.95 kb were generated using the PacBio platform (Supplementary Table 1). The filtered long reads were error-corrected, trimmed, and assembled, followed by filtering of the extra heterozygous fragments and the total length of the draft genome was 1.05 Gb, including 2035 contigs with an N50 length of 1.20 Mb (Supplementary Table 2). Next, 136.55 Gb of the high-throughput chromosome-conformation capture (Hi-C) data was generated and further mapped to the *A. elata* draft assembly (Supplementary Fig. 3). Finally, 99.12% of the assembled contigs, representing a 1.05-Gb genome, were anchored to 12 chromosome-level pseudomolecules (2n = 24), with a scaffold N50 value of 85.56 Mb (Supplementary Fig. 4; Supplementary Tables 3, 4). After mapping the Illumina reads to the final assembly, single-nucleotide polymorphisms (SNPs) were identified, and a SNP heterozygosity level of ~0.57% was obtained.

Based on the combined data from ab initio, homology-based analyses and RNA sequencing-assisted annotation, the *A. elata* genome was predicted to contain 35,042 protein-coding genes (Supplementary Table 5). Approximately 96.02% of the genes were functionally annotated by similarity searches against homologous sequences and protein domains (Supplementary Table 6). Repetitive elements account for 65.60% (686.33 Mb) of the *A. elata* genome (Supplementary Table 7); among them, long terminal- repeat (LTR) retrotransposons accounted for the largest proportion and made up 60.59% of the genome (Supplementary Tables 8, 9). The quality of the assembled genomes was then assessed with a series of analyses. First, a high LTR assembly index (LAI) score of 19.05 suggested that the continuity of the *A. elata* genome attained reference quality. Second, 99.52% of the Illumina reads were successfully mapped to the final genome, further supporting a high level of genome coverage. Third, a BUSCO (Benchmarking Universal Single-Copy Orthologs) assessment implied that a total of 98.80% and 98.10% of BUSCO gene models for genome assembly and predicted coding genes were respectively identified, suggesting the near completeness of genome assembly and annotation (Supplementary Table 10). All analyses suggested a high quality of the *A. elata* genome assembly and annotation.

**Phylogenetic analysis and whole-genome duplication events in *A. elata***. We constructed a phylogenetic tree and estimated the divergence times of 16 plant species using 145 single-copy genes (Fig. 2a; Supplementary Figs. 5, 6). As expected, *A. elata* clustered with three other Araliaceae species, *P. ginseng*, *P. notoginseng*, and *Eleutherococcus senticosus*. Molecular clock analysis indicated that *A. elata* diverged from *D. carota* (Apiaceae) at approximately 60.9 million years ago (Mya) (95% CI: 52.2–69.8 Mya) and from *Panax* of the *Aralia-Panax* monophyletic group approximately 12.3 Mya (95% CI: 7.4–17.7 Mya) (Fig. 2a; Supplementary Fig. 7). Evolutionary analysis of gene family expansion (i.e., gain) and contraction (i.e., loss) among candidate species (Supplementary Fig. 5 and Supplementary Table 11) showed that 510 gene

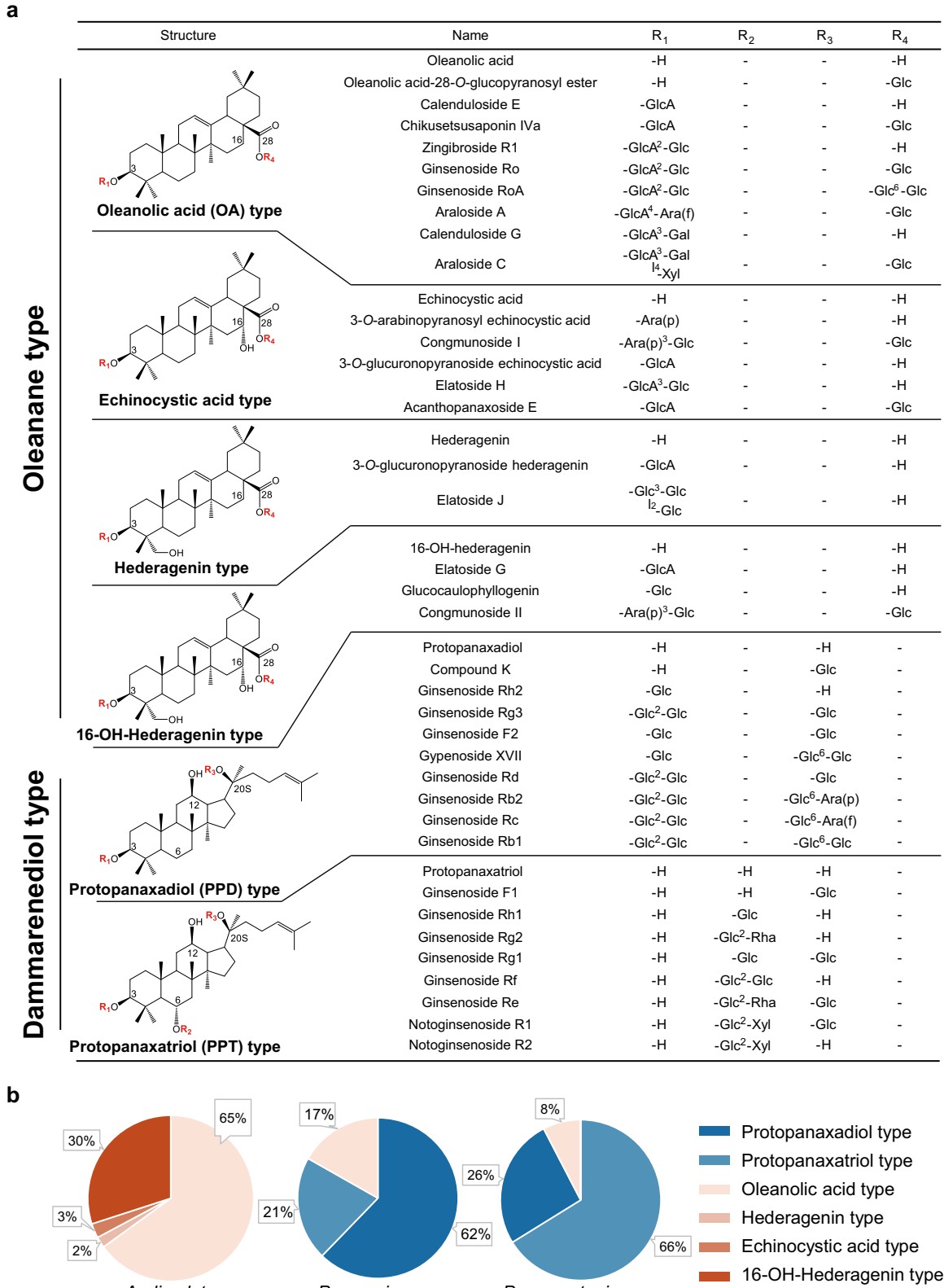

**Fig. 1 Structural and compositional variations in triterpenoid saponins among *Panax* and *Aralia* genera. a** Structures of different types of triterpene saponins isolated from *A. elata*, *P. ginseng*, and *P. notoginseng*. Glc D-glucopyranosyl, GlcA D-glucuronopyranosyl, Gal D-galactopyranosyl, Ara(p) L-arabinopyranosyl, Ara(f) L-arabinofuranosyl, Rha L-rhamnopyranosyl, Xyl D-xylopyranosyl. **b** Compositional variations in saponins with different triterpene skeletons in *A. elata*, *P. ginseng*, and *P. notoginseng*. The major triterpene profiles from a mixture of leaves, stems, and roots were determined using LC–MS. Source data underlying **b** are provided as a Source Data file.

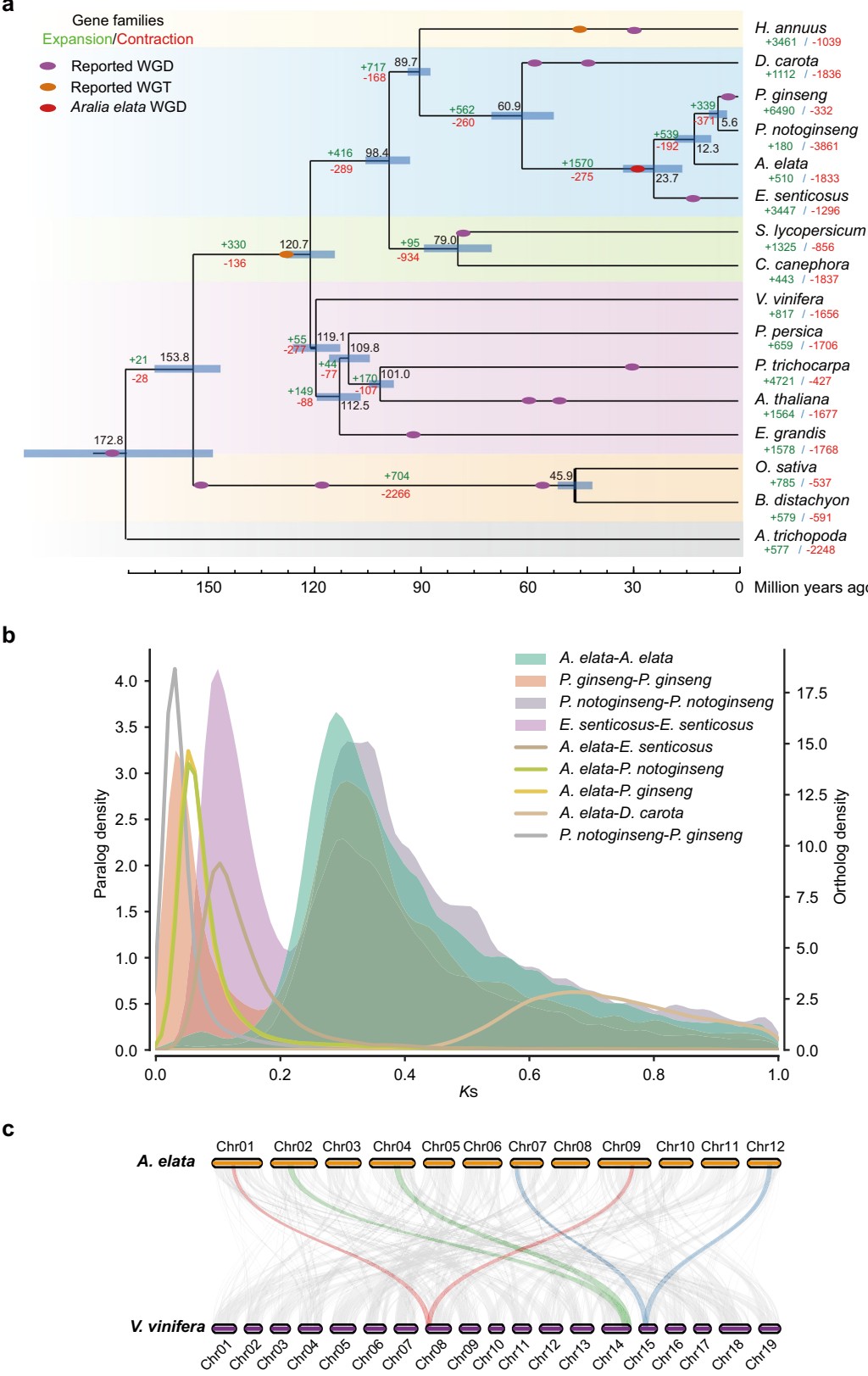

**Fig. 2 Genome evolution of *A. elata*. a** Inferred phylogenetic tree. Gene family expansions are indicated in green, and gene family contractions are indicated in red. The timings of whole-genome duplication (WGD) and whole-genome triplication (WGT) events are superimposed on the tree. Divergence times were estimated by MCMCTree, and the blue bars show the 95% confidence intervals of divergence times in millions of years. The black numbers near the nodes indicate the median divergence times. **b** Distributions of synonymous substitutions per synonymous site (*K*s) of one-to-one orthologs identified between *A. elata*, *P. ginseng*, *D. carota*, *E. senticosus*, and *P. notoginseng*. **c** Syntenic plots show a 2:1 syntenic relationship between the *A. elata* genome and *V. vinifera* genome.

families in the *A. elata* genome had expanded and 1833 gene families had contracted (Fig. 2a). Both Gene Ontology (GO) and Kyoto Encyclopedia of Genes and Genomes (KEGG) annotations showed that the expanded gene families in *A. elata* were significantly enriched in terpenoid biosynthetic pathways (Supplementary Fig. 8). Moreover, genes associated with response to wounding, regulation of defense response, and cellular response to cold were also enriched, indicating that the specific expansion of gene families related to these biological processes may play an important role in the environmental adaptability of *A. elata*.

Gene families usually expand by whole-genome duplication (WGD), segmental duplication, and tandem duplication[28]. Given that WGD is an important evolutionary force contributing to the diversity of specialized metabolites in plants[29], we investigated WGD events of *A. elata*. Synonymous substitutions per site ($Ks$) for paralogs of the *A. elata*, *E. senticosus*, *P. notoginseng,* and *P. ginseng* genomes showed a distinct peak at $Ks = 0.3$, which represents a known conserved WGD event across Araliaceae species[19–24]. In addition, the $Ks$ distribution of the orthologous genes between *A. elata* and *Panax* had a peak at $Ks = 0.07$. Therefore, the recent WGD event of *A. elata* might be shared with *P. notoginseng* and *P. ginseng*, occurring before the divergence of *Aralia* from *Panax* (Fig. 2b). Intergenomic collinearity analysis indicated a 2:1 collinearity relationship between *A. elata* and *V. vinifera* (Fig. 2c and Supplementary Fig. 9) and 1:1 and 1:2 collinearity relationships of *A. elata* with *P. notoginseng* and *E. senticosus*, respectively, which confirmed that all four Araliaceae species shared one WGD event, named the Araliaceae-common WGD[19,20] (Supplementary Figs. 10, 11). We further analyzed the genome synteny and $Ks$ values of paralogs and orthologs among *A. elata*, *D. carota*, and *V. vinifera* genomes. The results supported that the Dc-alpha and Dc-beta events are species-specific in *D. carota*, and the *A. elata* WGD event after γ event is shared by Araliaceae species after the split with *D. carota* (Supplementary Figs. 12–16, Supplementary Note 1). Furthermore, reconstruction analysis of the ancestral chromosomes indicated that the genomes of Araliaceae species were duplicated from a common ancestor with 13 chromosomes, despite extensive genome reorganizations after the WGD (Supplementary Fig. 17). In summary, no *Aralia*-specific WGD event was found in *A. elata*, indicating that compositional differences in the triterpenoid saponins between *A. elata* and *Panax* may not have derived from the WGD event.

**Partial deletion of *DDS* leads to the absence of tetracyclic triterpenes in *A. elata*.** Araliaceae plants are famous for richness of tetracyclic triterpene and pentacyclic triterpene saponins, and the composition and abundance of these metabolites are different among species (Fig. 1). GC–MS analysis showed the enriched accumulation of pentacyclic triterpenoids, including oleanolic acid, hederagenin, and their derivatives, in *P. ginseng*, *P. notoginseng*, and *A. elata*. Surprisingly, dammarane-type tetracyclic triterpene skeletons, such as protopanaxadiol, protopanaxatriol, and their derivatives, were observed in the roots and leaves of *P. ginseng* and *P. notoginseng,* but could not be detected in any tissue of *A. elata* (Fig. 3a and Supplementary Fig. 18).

The diversification between oleanane-type pentacyclic triterpenoids and dammarane-type tetracyclic triterpenes resulted from the divergent evolution of the oxidosqualene cyclases (OSCs); herein, we systematically identified *OSCs* in genomes of *A. elata*, *P. ginseng*, and *P. notoginseng*. Consistent with the distributions of pentacyclic triterpenoid derivatives, all three species contained β-amyrin synthase (*BAS*) genes, and many Araliaceae–*BAS* genes showed high expression in all the tested tissues of the three plants. In previous studies, dammarenediol synthases (DDSs) of the OSC

family in *P. ginseng* and *P. notoginseng* were found to contribute to the formation of tetracyclic triterpenes, and DDS-encoding genes showed high expression in the root, stem, and leaf tissues of *Panax* species. However, no *DDS* orthologs in the *A. elata* genome were identified (Fig. 3b, Supplementary Figs. 19, 20). Although the amino acid sequence similarity between AeAS3 (AE10G00915) and the CaDDS from *Centella asiatica* was 85%, the *AeAS3* gene was not expressed in any tissues of the *A. elata* (Fig. 3b; Supplementary Figs. 19, 20)[30]. Expression of recombinant AeAS3 in the yeast strain WAT21 resulted in the formation of a mixture of α- and β-amyrin, but no dammarenediol II was detected, indicating that AeAS3 is a multifunctional OSC producing α- and β-amyrin, but not a dammarenediol synthase (Supplementary Fig. 21).

We further identified syntenic regions containing *DDS* genes between the *A. elata* and *P. notoginseng* genomes via colinearity analysis. This DDS-syntenic block contains two tandem copies of *PnDDS* in chromosome 3 of *P. notoginseng* and an incomplete orthologous *DDS* with only 5 exons in *A. elata* (the *PnDDS* gene contains 17 exons) (Fig. 3c). Moreover, *E. senticosus* does not contain *DDSs* in homologous chromosomal regions (Supplementary Fig. 22). This suggested that the ancestral *DDS* gene underwent pseudogenization via the losing of multiexons after the speciation of *A. elata*. As the remaining incomplete *AeDDS* exons differentiated earlier than the *PnDDS* and *PgDDS* exons (Supplementary Fig. 23), we deduced that the dammarane-type OSCs are not neofunctionalized from other OSC copies but maintained by vertical inheritance during evolution. Furthermore, we overexpressed the *PgDDS* gene from *P. ginseng* in *A. elata* callus cells, and LC–MS analysis results showed that the overexpression of the *PgDDS* gene reconstructed the accumulation of dammarenediol II and its hydroxylated derivatives in the transgenic lines of *A. elata* (Fig. 3d, e). In addition, AeCYP716A357, the protopanaxadiol synthase ortholog in *A. elata*, was expressed ubiquitously in all the organs tested in *A. elata,* and was able to catalyze the formation of protopanaxadiol from dammarenediol II when it was expressed heterologously in yeast (Fig. 4b; Supplementary Figs. 24, 25). These results indicated that the tailoring enzymes, which catalyze the formation of dammarane-type tetracyclic triterpenes, are conserved between *A. elata* and *Panax* species. In summary, our results strongly supported that the deletion of 12 exons of the *DDS* gene during the evolution of the *A. elata* genome resulted in the loss of the ability to synthesize dammarane-type saponins.

**Functional identification of candidate genes responsible for oleanane-type triterpene biosynthesis in *A. elata*.** The synthesis of oleanane-type pentacyclic triterpenoids started from the cyclization of 2,3-oxidosqualene under the catalysis of β-amyrin synthase. To identify the β-amyrin synthase of *A. elata*, we identified 5 *OSC* genes in the *A. elata* genome (Fig. 3b and Supplementary Fig. 19). Then, we cloned the *AeBAS1* (AE05G00216) gene, which presents the highest expression level in leaves, and *AeBAS2* (AE04G00581), which shows the highest amino acid sequence similarity to AeBAS1. These two *AeOSCs* were transformed into yeast strain WAT21, which synthesizes 2,3-oxidosqualene[31], and the targeted β-amyrin was detected in both constructed strains (Fig. 4c).

Oleanolic acid, hederagenin, echinocystic acid, and 16-OH-hederagenin, the main pentacyclic triterpene skeletons of *A. elata*, were probably generated by oxidation at C28 and hydroxylation at C23 and C16 of β-amyrin sequentially (Fig. 4a). These reactions are mainly catalyzed by the P450 superfamily, especially the members of the CYP716A and CYP72A subfamilies[17,32]. To select

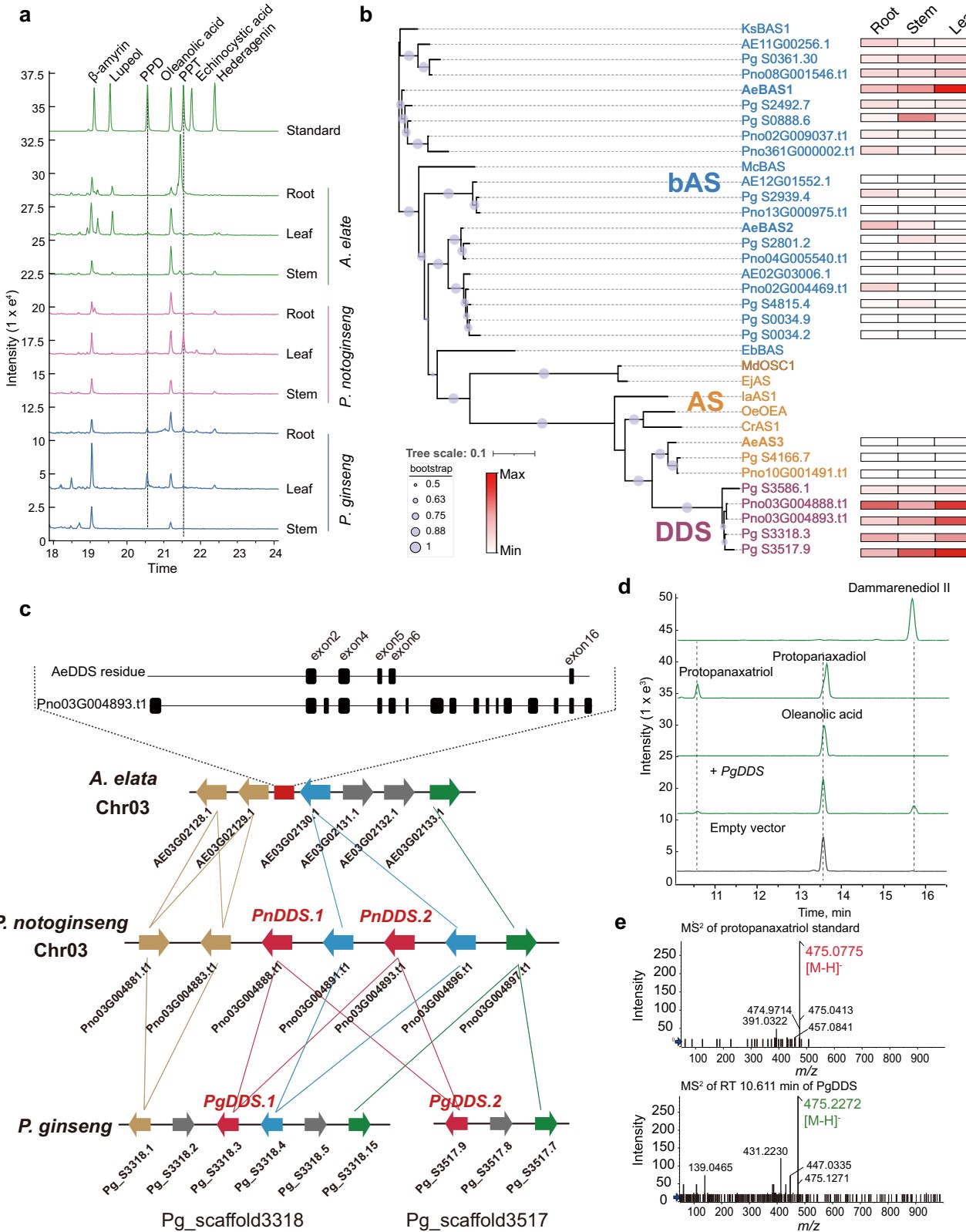

**Fig. 3 The loss of *DDS* gene in the genome of *A. elata*. a** GC–MS analysis of triterpenoid contents in *A. elata*, *P. notoginseng*, and *P. ginseng*. **b** Phylogenetic tree of BAS, AS, and DDS family proteins in *P. ginseng*, *P. notoginseng,* and *A. elata*. The phylogenetic tree was constructed using the maximum likelihood method, in which tree nodes (>50%) are indicated by purple dots on branches (1000 bootstrap replications). The heatmap on the right shows the FPKM values of gene expression in the roots, stems, and leaves of *A. elata*. **c** Syntenic analysis of *DDS* genes in *P. ginseng*, *P. notoginseng*, and *A. elata*. The red rectangle indicates the position of DDS residues, the rounded rectangles in the dashed box indicate exons, the lines indicate genes with colinearity, and the red lines indicate the colinear relationships of *DDS* genes between *P. notoginseng* and *P. ginseng*. **d**, **e** LC–MS analysis of triterpenoid contents of *Aralia elata* callus overexpressing *PgDDS*. Source data underlying **b** are provided as a Source Data file.

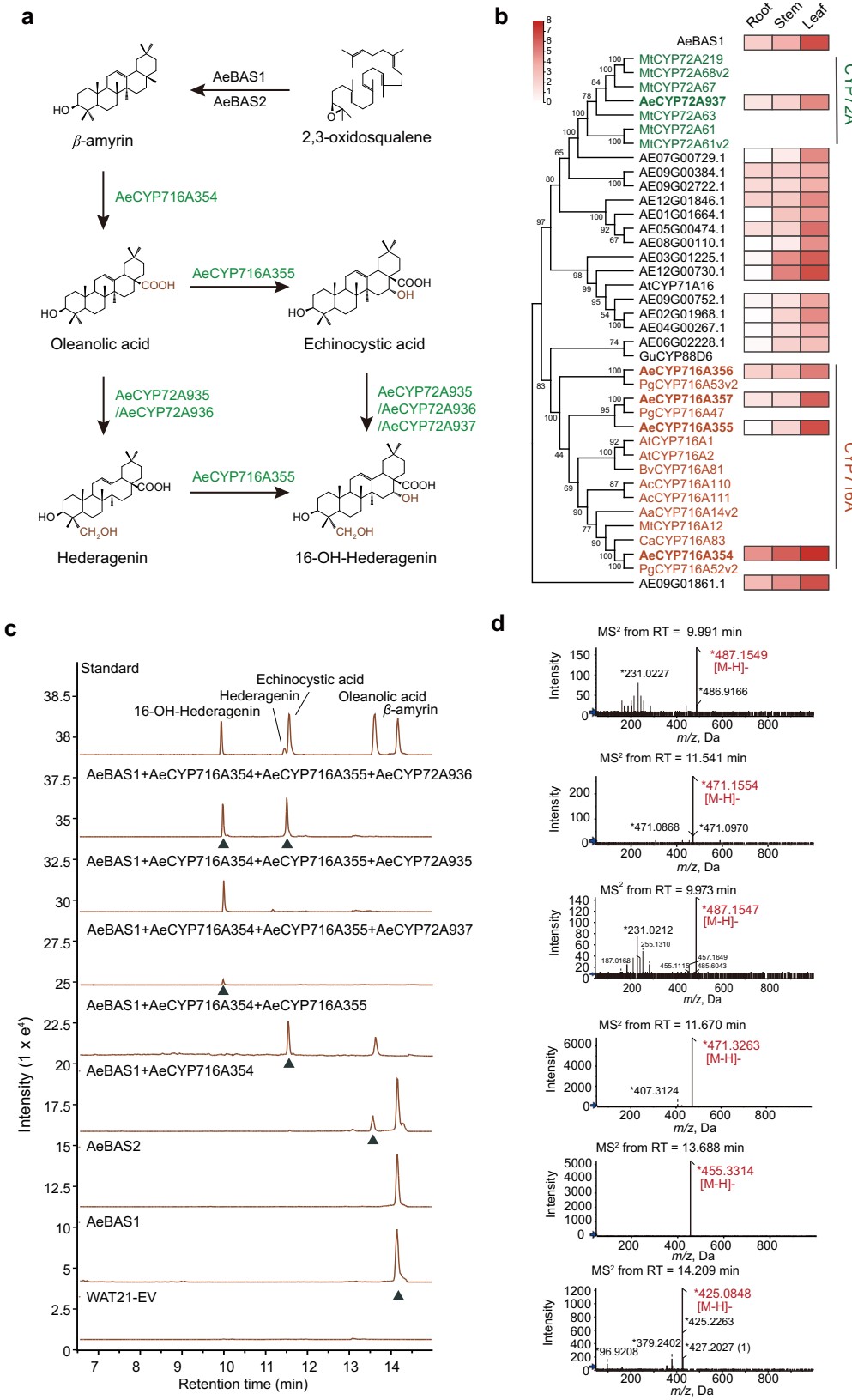

the key P450s related to the biosynthesis of oleanane-type saponin backbones, we conducted co-expression analyses between the 256 *P450* genes annotated in the *A. elata* genome and the β-amyrin synthase gene *AeBAS1* (Supplementary Figs. 24, 26). Among them, the expression patterns of 18 *P450* genes were similar to that

of *AeBAS1*, and phylogenetic analysis showed that four *P450* members (*AeCYP716A354*, *AeCYP716A355*, *AeCYP716A356*, and *AeCYP716A357*) belong to the *CYP716A* subfamily and one *P450* gene from the *CYP72A* subfamily (*AeCYP72A937*) in *A. elata* (Fig. 4b and Supplementary Fig. 24).

**Fig. 4 Hydroxylation process of the oleanane-type skeletons in *A. elata*. a** CYP-catalyzed hydroxylation of oleanane-type triterpenoids. **b** The heatmap highlights the patterns of coexpression of *P450* genes with *AeBAS1* in the root, stem, and leaf tissues of *A. elata*, with FPKM values used for normalization and color-coding conducted accordingly. The phylogenetic tree of these P450s and other P450s characterized in *Panax ginseng* (Pg), *Arabidopsis thaliana* (At), *Medicago truncatula* (Mt), *Barbarea vulgaris* (Bv), *Glycyrrhiza uralensis* (Gu), *Artemisia carvifolia* (Ac), *Artemisia annua* (Aa), and *Centella asiatica* (Ca). The CYP716A and CYP72A subfamily members are marked in red and green, respectively. **c** LC–MS indicates that AeBAS1, AeBAS2, AeCYP716A354, AeCYP716A355, AeCYP72A935, and AeCYP72A936 are functional in yeast. Extract ion chromatogram overlay of the strains expressing the target genes or the empty vector control (WAT21-EV). Peaks potentially corresponding to saponins are labeled with the *m/z* value of β-amyrin (*m/z* 426), oleanolic acid (*m/z* 455), echinocystic acid (*m/z* 471), hederagenin (*m/z* 471), and 16-OH-hederagenin (*m/z* 487). The new peak of engineered strains is marked by a green triangle, and the MS/MS spectra are shown in **d**. The observed fragmentation of each spectrum is marked in red. Source data underlying **b** are provided as a Source Data file.

The functional identification of these five CYPs indicated that three of them were involved in the synthesis of oleanane-type saponin skeletons. AeCYP716A354 (AE01G03625) could catalyze oxidation at the C28 position of β-amyrin, leading to the formation of oleanolic acid, and AeCYP716A355 (AE10G01443) could hydroxylate the C16 position of oleanolic acid to produce echinocystic acid. AeCYP72A937 (AE09G02158), a CYP72A subfamily member, could hydroxylate the C23 position of the hederagenin to form the 16-OH-hederagenin (Fig. 4c, d).

**Tandem gene duplications lead to the diversity of oleanane-type triterpenoids.** The pentacyclic triterpenoids in *A. elata* are more abundant and diverse than those in *P. ginseng* and *P. notoginseng*. Oleanane-type saponins are the main pentacyclic triterpenoids in *P. ginseng* and *P. notoginseng*, while *A. elata* is also enriched in oleanane-type, hederagenin-type, and echinocystic acid-type saponins and their glycosylated derivatives (Fig. 1). As previously reported, P450s, CSLs, and UGTs are the key tailoring enzymes in the later steps of the triterpene saponin biosynthetic pathway[18,33]. In the *A. elata* genome, we could not find that the above pentacyclic triterpenoid biosynthetic genes formed any metabolic synthetic gene cluster; instead, the P450, CSL, and UGT gene families frequently appeared as tandem duplications spread along the genome (Supplementary Fig. 27).

For example, two tandemly duplicated CYP72A subfamily members with a sequence similarity over 90% (Supplementary Fig. 28), *AeCYP72A935* (AE09G00610) and *AeCYP72A936* (AE09G00613), were located on chromosome 9 of *A. elata* within 130 kb (Fig. 5a). Phylogenetic and *K*s analyses suggested that the *Aralia*-specific gene duplication event of AeCYP72A occurred approximately 4.51 Mya after the *Aralia-Panax* divergence (Supplementary Fig. 29a). Functional identification shows that these two CYP72A family proteins possess the same catalytic capacity, hydroxylating oleanolic acid and echinocystic acid into hederagenin and 16-OH-hederagenin, respectively (Fig. 4c). After purification, the structure of 16-OH-hederagenin was identified using ${}^1$H NMR and ${}^{13}$C NMR (Supplementary Fig. 30)[34]. *AeCYP72A935* and *AeCYP72A936*, which encoded proteins responsible for hydroxylation at the C23 position of the pentacyclic triterpene, are mainly expressed in stems. In contrast, the oleanolic acid synthase gene *AeCYP716A354* and echinocystic acid synthase gene *AeCYP716A355* had the highest expression levels in the leaves.

Recently, cellulose synthase-like M-subfamily (CSLM) enzymes were reported to catalyze the 3-*O*-glucuronosylation of triterpenoid aglycones[18,33]. A tandem duplicate cluster of 3 tightly linked *CSLMs* on chromosome 6 of *A. elata* was identified (Fig. 5b; Supplementary Figs. 31, 32). In contrast, there was only one orthologous *CLSM* in the colinear region of *P. notoginseng*. Phylogenetic and *K*s analyses showed that tandem duplication of *AeCSLM* occurred approximately 12.34 Mya before, that is, after the species differentiation of *A. elata* and *P. notoginseng* (Supplementary Fig. 29b). Functional identification showed that products of the two adjacent CSLMs that had different expression

patterns, *AeCSLM1* (AE06G00237) and *AeCSLM2* (AE06G00238), can perform the same activities (Fig. 5b and Supplementary Fig. 33), i.e., transfer a glucuronic acid sugar group to the C3 carbon of four different skeletons of oleanane-type triterpenes.

The plant UGT73 family is involved in the glycosylation of oleanane-type pentacyclic triterpenes[25], and many UGT73 family members were found in the *A. elata* genome as homologous gene clusters. For example, we found a tandem repeat including a five-gene cluster located on chromosome 9 within a 35-kb region (Fig. 5c and Supplementary Fig. 34). The duplication events of *AeUGT73CB4* (AE09G01562 and AE09G01563), and *AeUGT73CB5* (AE09G03873), among these daughter genes, may have happened after the divergence of *A. elata* and *P. notoginseng* (Supplementary Fig. 29c). The pattern of *AeUGT73CB3* (AE09G01544) expression at a high level in the leaves differed from *AeUGT73CB4* and *AeUGT73CB5*. In vitro functional analysis showed that *AeUGT73CB3*, *AeUGT73CB4*, and *AeUGT73CB5*, which encode the enzymes, add a glucosyl to the C3 position of calenduloside E to generate zingibroside R1 through the formation of a $(1 \rightarrow 2)$-glycosidic bond. However, the turnover rates of AeUGT73CB4 and AeUGT73CB5 for calenduloside E were lower than 52% of that of AeUGT73CB3, and the ancestral gene products of *AeUGT73CB2* (AE09G01543) did not exhibit the same glycosylation activity (Supplementary Fig. 35). The kinetic parameters of AeUGT73CB3 for calenduloside E and chikusetsusaponin IVa as substrates were analyzed. AeUGT73CB3 had a $K_m$ value for calenduloside E that was 12.5-fold lower than that for chikusetsusaponin IVa, and a $k_{cat}$ value for calenduloside E that was twofold higher than that for chikusetsusaponin IVa (Supplementary Table 12), indicating AeUGT73CB3's relatively high affinity to calenduloside E and decent efficiency in catalyzing the 3-*O*-glucosylation of calenduloside E.

Moreover, another six UGT73 family members were also localized as a tandem gene array in a 40-kb region of chromosome 9 of *A. elata*. Syntenic analysis revealed that this gene cluster is highly conserved in *P. ginseng* and *P. notoginseng*, indicating that it originated before three genera separated (Fig. 5d; Supplementary Fig. 29d). Only AeUGT73AD2 (AE09G02726) was found to exhibit glycosylation activity at the C28 position of the four pentacyclic triterpene skeletons after in vitro functional identification (Supplementary Fig. 36), while the other UGTs were not found to be involved in pentacyclic triterpene synthesis.

In addition, we also tested the enzymatic activity of the Pno09G000253, Pno06G01541, Pno14G001407, and Pno14G001424 from *P. notoginseng*, which located at the colinear regions corresponding to the tandem repeats of tailoring enzymes in *A. elata*. The results showed that these colinear genes in *P. notoginseng* possess similar functions for hydroxylation or glycosylation as their orthologs in *A. elata* (Supplementary Fig. 37). However, the expression levels of the orthologous genes of *AeCYP72A935* and *AeCYP716A355* in *P. notoginseng* were extremely low in all the tissues tested, which may limit the accumulation of oleanane-type pentacyclic triterpenes. In summary, *A. elata* oleanane-

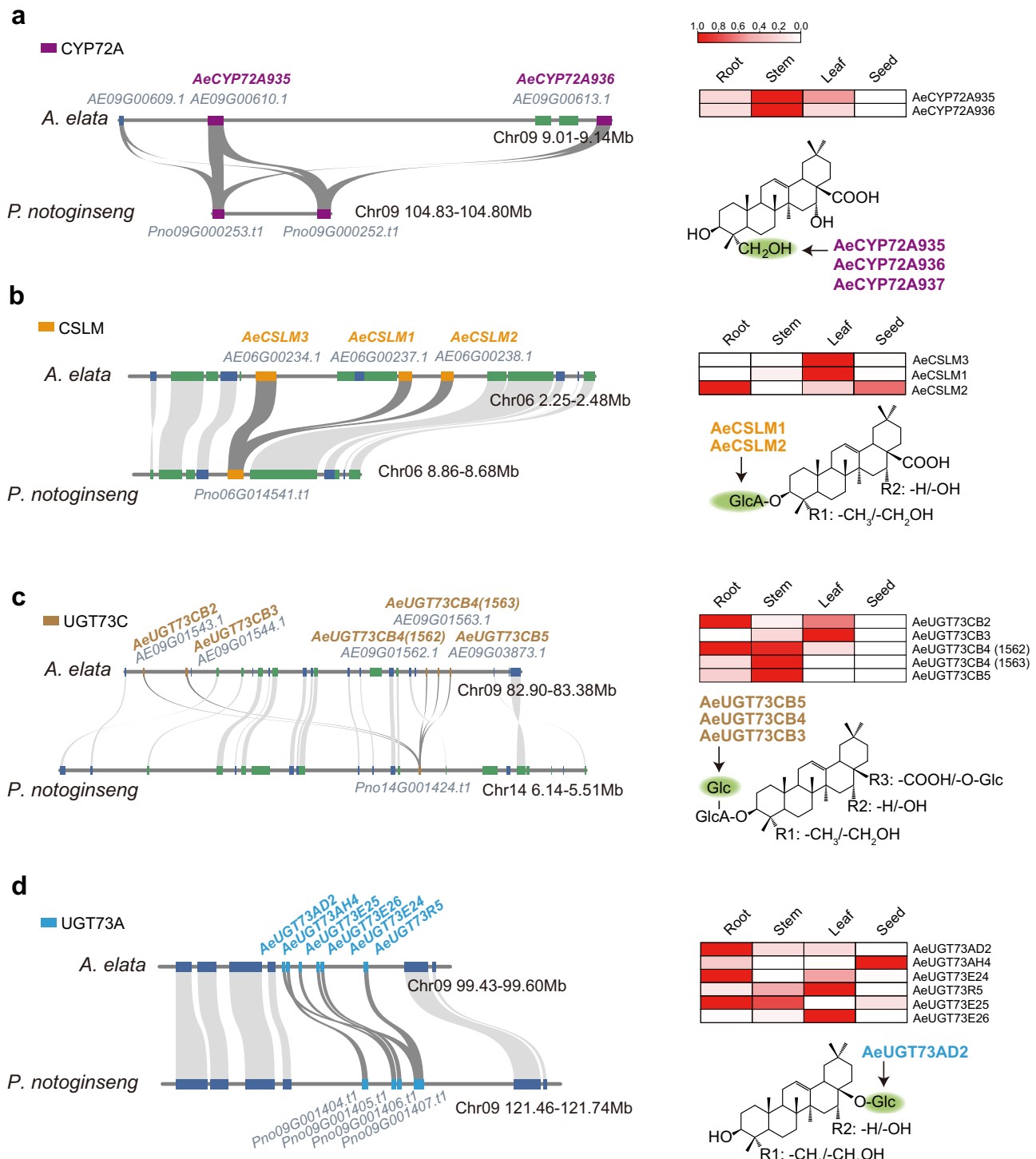

**Fig. 5 Tandem duplication of genes drives the diversity of oleanane-type pentacyclic triterpenes in *A. elata*.** The colinearity relationship of CYP72A (**a**), CSLM (**b**), and two UGT73 (**c** and **d**) tandem duplication regions between *A. elata* and *P. notoginseng*. The syntenic blocks are connected by gray lines. The heatmap shows the FPKM values (normalized) of the colinear gene in different tissues of *A. elata*. The green circles represent the tandem-repeat enzyme-modification positions. The blue bar represents the genes on the forward strand of the genome; the green bar represents genes on the reverse strand. Source data are provided as a Source Data file.

type pentacyclic triterpene biosynthetic pathways have diverged due to tandem gene duplications and various expression patterns of *P450*, *CSL*, and *UGT73A* genes.

**De novo biosynthesis of oleanane-type pentacyclic triterpenes in yeast.** In *A. elata*, the C28 glycosylated saponins (ginsenoside Ro and chikusetsusaponin IVa) were abundant, notably in roots (Supplementary Fig. 18). We speculated that another UGT catalyzes the C28 position glycosylation of calenduloside E in the roots since AeUGT73AD2 lacks the glycosylation activity of calenduloside E and zingibroside R1 (two oleanane-type pentacyclic triterpenes of C3 glucuronidation). Then, we screened the *UGTs* that were specifically expressed at high levels in roots

(Supplementary Fig. 38, 39); among them, the enzyme encoded by AeUGT74AG6 (AE01G00407) exhibited C28 glycosylation activity for oleanane-type pentacyclic triterpenes of C3 glucuronidation (Supplementary Fig. 40), which may compensate the glycosyltransferase requirement at position C28 in A. elata roots. The AeUGT73AD2 showed a $K_m$ value to oleanolic acid close to that of AeUGT74AG6, but the latter could catalyze two new substrates, calenduloside E and zingibroside R1. AeUGT74AG6 exhibited low $K_m$ values (implicating high substrate specificity) to zingibroside R1 and oleanolic acid, and high catalytic efficiency (based on the $k_{cat}/K_m$ value) to zingibroside R1, when compared with its parameters to calenduloside E (Supplementary Table 12).

To de novo produce diverse oleanane-type pentacyclic triterpenes, we introduced the key enzymes involved in the biosynthetic pathway of pentacyclic triterpenoids into Saccharomyces cerevisiae. Yeast cells naturally provide endogenous 2,3-oxidosqualene, the archetypical substrate of β-amyrin synthases. Moreover, an AtATR2 construct in yeast strain WAT21 provides reducing equivalents from NADPH to P450. Here, we enabled overexpression of the genes from the MVA pathway in WAT21 to generate yeast strain WOA and fully integrated β-amyrin synthase and 2 P450 genes, AeCYP716A354 and AeCYP716A355, into strain WOA to generate strain WEA (Fig. 6a and Supplementary Fig. 41). Then, after AeCYP72A936 was inserted into strain WEA, we detected four types of pentacyclic triterpene skeletons, oleanolic acid, hederagenin, echinocystic acid, and 16-OH-hederagenin, in the corresponding strains (Fig. 6b). Because yeast cells lack the sugar donor UDP-glucuronic acid for the C3 glucuronosylation reaction of pentacyclic triterpene skeletons, UDP-glucose 6-dehydrogenase 1 (UGD1) from E. coli was expressed in yeast to enable the synthesis of UDP-glucuronic acid. The constructed strains that expressed EcUGD1 and AeCSLM1 in strain WEA synthesized two C3-glucuronated aralosides: calenduloside E (m/z 631) and echinocystic acid 3-O-glucuronopyranoside (m/z 647) (Fig. 6a, b; Supplementary Fig. 42). Furthermore, AeUGT73CB3 glucosylates the C3 position of these two C3-glucuronated oleanane-type pentacyclic triterpenes to generate two other kinds of oleanane-type pentacyclic triterpenes in yeast (compounds 8 and 13). Finally, the addition of glucose at the C28 position via glucosyltransferase AeUGT74AG6 leads to the production of various glycosylation products, including the highest-glucosylated saponins (compounds 9 and 14) and intermediate metabolites (compounds 6, 7, and 12) (Fig. 7). Based on the molecular weights measured in mass spectrometry and the enzymatic characteristics of these glycosyltransferases, the structure of compounds 13 and 14 is inferred to be echinocystic acid 28-O-glucopyranosyl-3-O-glucuronopyranoside and echinocystic acid 3-O-glucuronopyranosyl-6 → 1-glucoside, respectively (Supplementary Fig. 43).

Through the combination of different enzymes, we were able to synthesize 13 kinds of oleanane-type pentacyclic triterpenes, including high-content saponins (compounds 8 and 9) and low-content saponins (compounds 4 and 7) in A. elata (Supplementary Fig. 18). We inferred that β-amyrin can be continuously hydroxylated by different P450s to form a variety of skeletons, and then different types of glycosyltransferases are combined with P450s to further advance the structural diversity of pentacyclic triterpene saponins. Most importantly, the catalytic promiscuity of glycosyltransferases makes them able to produce multiple substrates. Araloside C is well known for its special physiological activity and is unique to A. elata[5]. We predict that downstream of the synthetic pathway of chikusetsusaponin IVa, araloside C may be synthesized using two other unique UGTs in A. elata (Fig. 7).

This study uncovered the genome of A. elata, a sister plant of Panax species, and revealed the divergent evolution of oleanane-type pentacyclic triterpenes and dammarane-type tetracyclic triterpenes between the closely related species A. elata and

Panax, which originated from the partial deletion of the DDS gene and tandem duplications of biosynthetic genes, including CSL, P450, and UGT genes, in the A. elata genome. The diversity of pentacyclic triterpenes and the illumination of their biosynthetic genes in A. elata provide important insights for its environmental adaptation and in vitro bioproduction of specific metabolites (Fig. 8 and Supplementary Fig. 44).

## Methods

**Plant materials and genome-survey analysis.** The experimental materials from a 5-year-old Aralia elata individual were harvested at Harbin, Heilongjiang Province, China. High-quality genomic DNA was extracted from the fresh leaf tissues of A. elata and used to construct libraries for Illumina, PacBio, and Hi-C sequencing at the Beijing Genomics Institute-Shenzhen (BGI-Shenzhen).

In all, 54.01-gigabase (Gb) Illumina paired-end clean reads were generated from the Illumina HiSeq 4000 system (Illumina, San Diego, CA) and filtered by SOAPnuke software v2.0.2[35] (filtration parameter: -n 0.02 -l 20 -q 0.4 -i -G 2 -seqType 1 -Q 2). We performed the investigation of genome survey by K-mer distribution analysis (K = 17) with Jellyfish[36] (v2.2.6) and GenomeScope[37] software to predict the genome size, heterozygosity, and repeat-sequence features.

**Genome sequencing, assembly, and quality assessment.** For PacBio libraries, the whole genome was sequenced on the PacBio Sequel II System based on single-molecule real-time (SMRT) sequencing technology, and 99.88 Gb (~90.6×) of data were obtained. The subreads obtained from PacBio libraries were assembled into contigs using Canu (v2.1)[38], and the consensus genome was polished by referring to the Illumina reads with BWA (v0.7.17)[39] and two rounds of corrections by Pilon (v1.23)[40]. The high-quality Hi-C data were used to further assist in chromosome-level genome assembly. We obtained 136.55 Gb of raw data, which were first filtered using SOAPnuke v2.0.2 with the following parameters: filter -n 0.01 -l 20 -q 0.4 -d -M 3 -A 0.3 -Q 2 -i. Juicer v2.0[41] was applied to align the clean data to the genome, and then duplicated and abnormal alignments were removed. The locations and directions of the contigs were preliminarily determined by 3d-DNA (v 180922)[42] with default parameters. To prevent excessive interruption, the result of the first iteration of 3d-DNA was used as input for Juicebox (v1.11.08)[43]. We visualized the Hi-C contact map and performed extensive manual curation by Juicebox to fix misjoins and clustered contigs.

We further used BUSCO v3 (embrophyta_odb10)[44] to evaluate the completeness of the genome assembly. For LAI analysis, we identified the LTR sequence in the Aralia elata genome using LTRdigest (options -minlenltr 100 -maxlenltr 7000 -mintsd 4 -maxtsd 6 -motif TGCA -motifmis 1 -similar 85 -vic 10 -seed 20) and LTR_finder (options -D 15000 -d 1000 -L 7000 -l 100 -p 20 -C -M 0.85). Next, we integrated the results from these two programs to calculate the LAI value using LTR_retriever[45]. For heterozygosity analysis, we identified SNPs using GATK with 'gatk -java-options -Xmx4G HaplotypeCaller' and extracted SNP information with 'gatk SelectVariants -select-type-to-include SNP'[46]. We filtered low-quality SNP sites using 'gatk VariantFiltration -filter-expression 'QUAL < 30.0 || QD < 2.0 || FS > 60.0 || SOR > 4.0' -filter-name lowQualFilter -cluster-window-size 10 -cluster-size 3 -missing-values-evaluate-as-failing'. Finally, we calculated the heterozygosity of SNP site that had a sequencing depth greater than 5.

**Genome annotation.** We combined three strategies to predict genes in the A. elata genome: homology-based, de novo, and RNA-Seq data alignment. For homology-based annotation, the protein sequences of P. ginseng, P. notoginseng, E. senticosus, Apium graveolens, Coriandrum sativum, Mikania micrantha, and Arabidopsis thaliana were downloaded from public repositories. These proteins were aligned to the A. elata genome using BLASTX[47] with a score threshold of '-e 1e−5'. Next, GeneWise v2.4.1[48] was applied to predict gene structure and to define gene models.

For de novo prediction, we randomly selected 5,000 full-length intact genes of A. elata (intact structure: start codon, stop codon, and perfect intron–exon boundary) from the gene models predicted by homology-based methods to train the model parameters for AUGUSTUS v3.2.1[49] and SNAP v2006-07-28[50]. Then, the two trained software programs were used to perform de novo prediction on the repeat-masked genome of A. elata.

For transcriptome-based prediction, RNA-Seq data were mapped onto the A. elata genome to identify splice junctions using HISAT2 v2.1.0 software[51]. StringTie v1.2.2[52] was then used to assemble transcripts with the aligned reads. Finally, the integrated consensus gene models were derived from MAKER v3.31.8[53] with upper AUGUSTUS and SNAP de novo prediction, the predicted gene structures derived from the six protein-based homologs, and RNA-Seq-based transcript structures.

Gene-function annotation was based on sequence similarity and domain conservation. First, the A. elata protein-coding genes were compared against the KEGG[54], SwissProt[55], TrEMBL[55], NR, and KOG[56] databases using BLASTP at a cutoff E-value threshold of $10^{-5}$. Subsequently, the best match from the alignment was used to represent the gene function. Second, InterProScan software v5.16–55.0[57] was used to identify the motifs and domains based on public

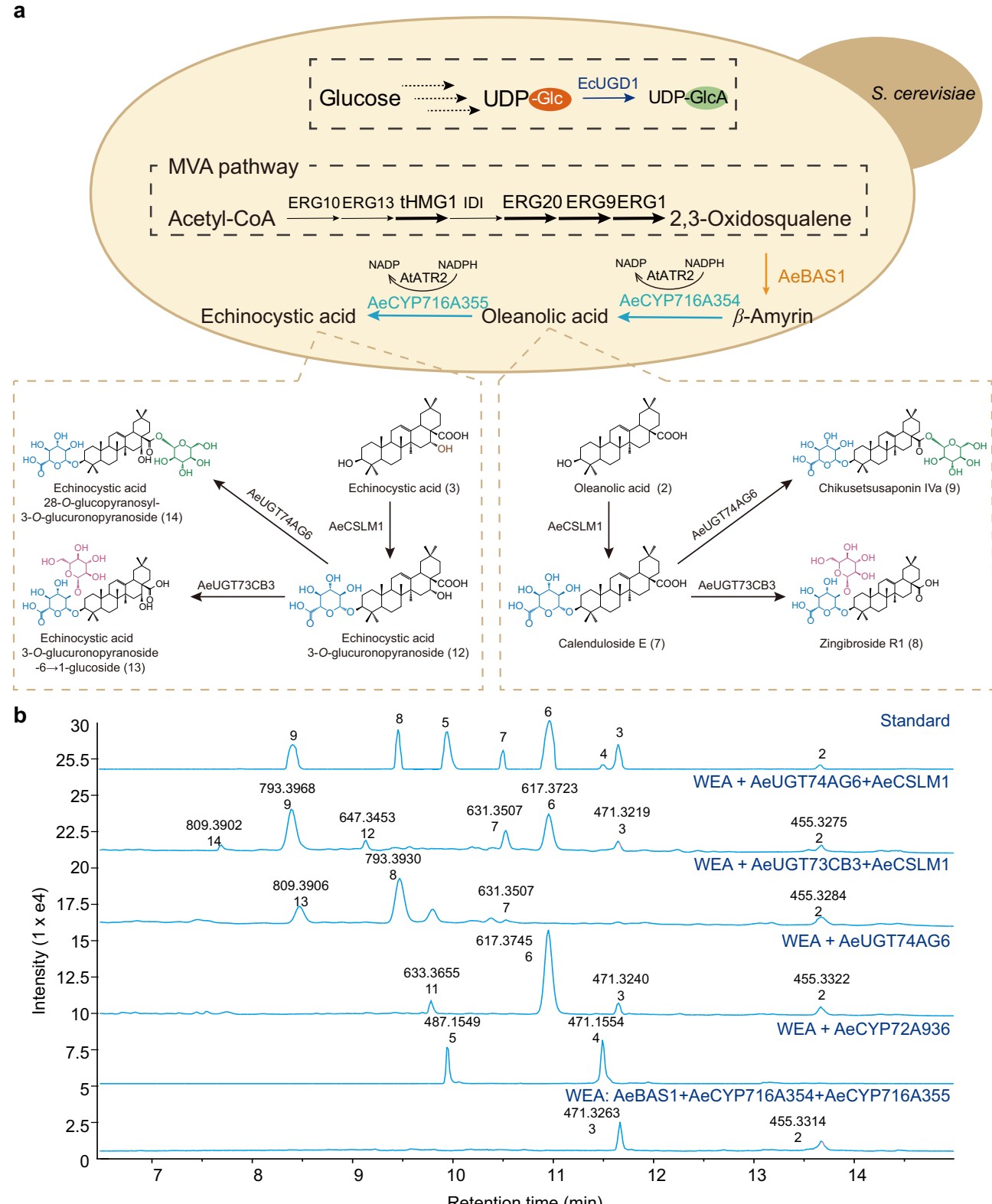

**Fig. 6 De novo biosynthesis of oleanane-type pentacyclic triterpenes in yeast. a** The strain WEA expressed *AeBAS*, *AeCYP716A354*, and *AeCYP716A355* and overexpressed the MVA and early sterol-pathway genes *tHMG1*, *ERG20*, *ERG9*, and *ERG1*. Diagram showing the echinocystic acid-type triterpenoid biosynthetic pathway. **b** LC–MS analysis of metabolites from strains engineered for pentacyclic triterpene saponin production derived from β-amyrin and oleanolic acid. Extracted ion chromatograms using *m/z* values (±0.5) and different numbers to represent saponins with various aralosides, oleanolic acid (*m/z* 455, 2), echinocystic acid (*m/z* 471, 3), hederagenin (*m/z* 471, 4), 16-OH-hederagenin (*m/z* 487, 5), oleanolic acid 28-*O*-glucopyranosyl ester (*m/z* 617, 6), calenduloside E (*m/z* 631, 7), zingibroside R1 (*m/z* 793, 8), chikusetsusaponin IVa (*m/z* 793, 9), echinocystic acid 28-*O*-glucopyranosyl ester (*m/z* 633, 11), 3-*O*-glucuronopyranoside echinocystic acid (*m/z* 647, 12), echinocystic acid 28-*O*-glucopyranosyl-3-*O*-glucuronopyranoside (*m/z* 809, 14), and echinocystic acid 3-*O*-glucuronopyranoside-6 → 1-glucoside (*m/z* 809, 13).

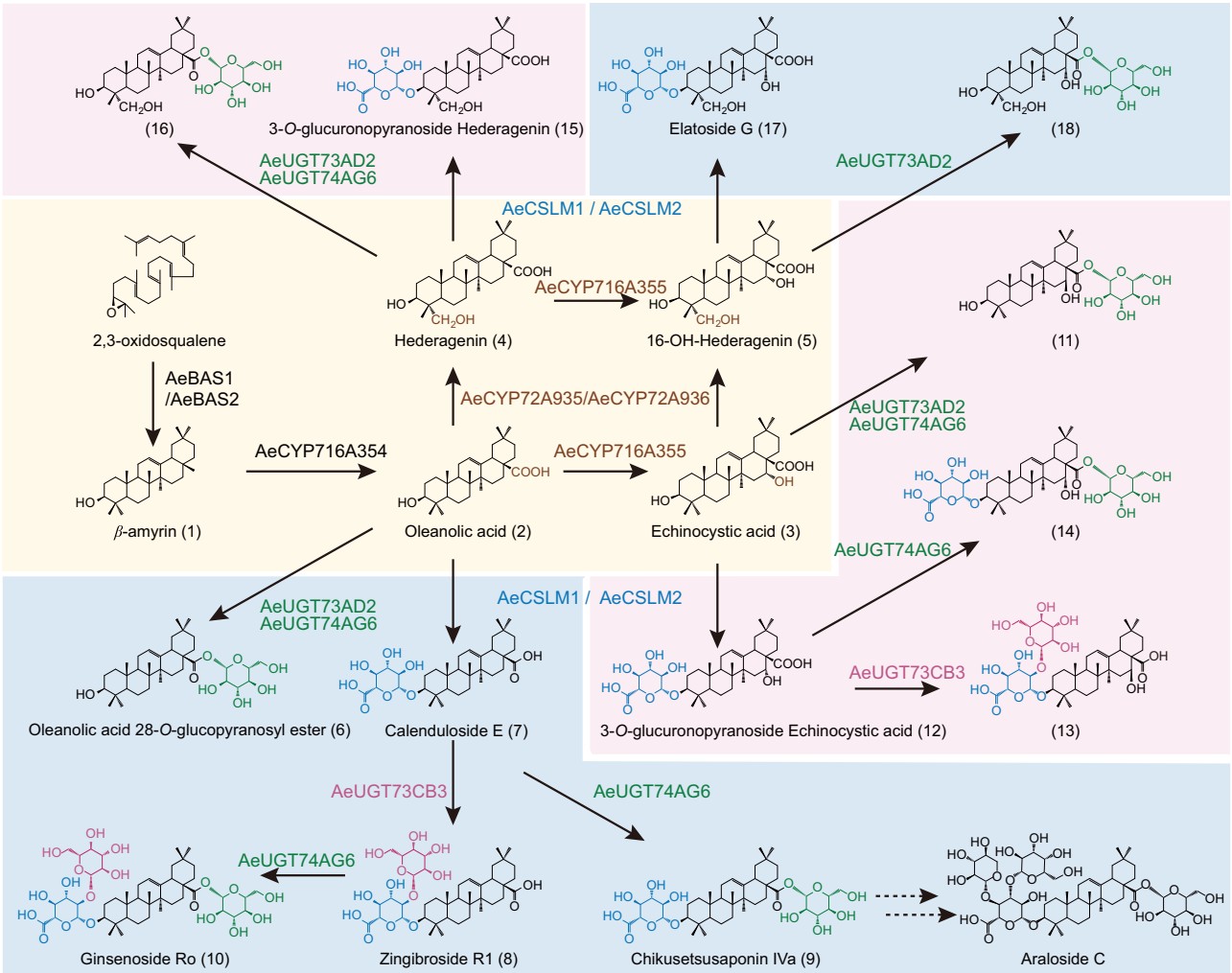

**Fig. 7 Overview of araloside structures and their biosynthesis in *A. elata*.** The light-yellow box indicates the cyclization of 2,3-oxidosqualene to β-amyrin and CYP-catalyzed hydroxylation of triterpenoids. The boxes in other colors represent the glycosylation of different triterpene backbones. Brown letters indicate hydroxylation reactions catalyzed by cytochrome P450 monooxygenases (P450s); blue letters indicate glycosylation reactions catalyzed by cellulose synthase-like glycosyltransferase (CSL); green letters indicate C28 glycosylation reactions catalyzed by UDP-dependent glycosyltransferases (UGTs); and purple letters indicate C3–2 glycosylation reactions catalyzed by UGTs. Reaction sites are indicated in the corresponding colors. Broken arrows indicate putative araloside biosynthesis steps.

databases. Gene Ontology (GO)[58] functional information was retrieved from InterProScan by searching GO terms with the parameter '-goterms'.

We combined homology-based annotation and a de novo method to identify transposable elements (TEs) and tandem repeats in the *A. elata* genome. In the homology-based annotation step, TEs were identified by searching against the Repbase v21.12 library with RepeatMasker v4.0.7[59] and RepeatProteinMasker v4.0.7[59]. During the de novo step, de novo TE libraries were constructed based on the genome sequences using the de novo prediction program RepeatModeler v1.0.11[60]. Subsequently, TEs were identified and classified using RepeatMasker. Tandem repeat sequences were identified by Tandem Repeat Finder v4.09 software[61] with the following parameters: 'Match = 2, Mismatch = 7, Delta = 7, PM = 80, PI = 10, Minscore = 50 and MaxPerid = 2000'.

**Genome evolution.** In this study, *A. elata* and 15 other species were used for gene family construction. For genes with alternative splicing variants, the longest transcripts were retained in these analyses. Similarities between protein sequences were calculated using BLASTP with an E-value threshold of 1e − 5, and then OrthoMCL v1.4[62] was used to identify the gene family based on the similarities of the genes and Markov chain clustering by setting the main inflation value at 1.5 and using the default settings for other parameters. Based on the identified gene families and constructed phylogenetic tree with the predicted divergence times of the 15 species, gene family expansion and contraction along the branches of the phylogenetic tree were analyzed using CAFE v2.1[63]. First, families with differences in size were discarded (e.g., families with ≥200 genes in one species and ≤2 genes in all other species), and families with a most recent common ancestor (MRCA) size

equal to 0 as predicted using a parsimony method were also filtered. In CAFE, the random birth-and-death model was proposed to study gene gain and loss in gene families across the specified phylogenetic tree. The R package ClusterProfiler v4.0[64] was used for GO and KEGG enrichment analyses of expanded and contracted gene families.

We extracted 145 single-copy orthologous genes derived from the single-copy gene family analysis and then aligned the protein sequences of each family using MUSCLE v3.8.31[65]. Next, the protein alignments were converted to the corresponding CDS using an in-house Perl script. Each amino acid was substituted with the corresponding triplet bases from its CDS according to the same ID information. These coding sequences of each single-copy gene family were concatenated to form one "supergene" for each species. The nucleotides at position two (phase-one site) of each codon were extracted separately to construct the phylogenetic tree using RAxML based on the GTRGAMMA model. The tree constructed using phase-one sites was consistent with that constructed with fourfold degenerate sites. The $K_s$ value of gene pairs was calculated using TBtools (v1.0987) with the Simple $K_a$/$K_s$ Calculator (NG) function[67].

We used MCSCAN python-version (v1.1.18)[68] to identify the synteny between *A. elata* and other species. For synteny within areca palm, we performed pairwise alignments using BLASTP v2.2.26 (E-value <1e−5). MCSCAN was used to construct syntenic blocks (parameters: -a -e 1e-5 -s 5). We calculated the $K_s$ of syntenic blocks to identify WGD events.

To infer the evolutionary relationships of Araliaceae, we first used BLAST to align the CDS and performed a filtering step on the basis of the cumulative identity percentage (CIP) and cumulative alignment-length percentage (CALP) to identify orthologs; then, we used MCSCAN software to analyze the orthologs that existed in

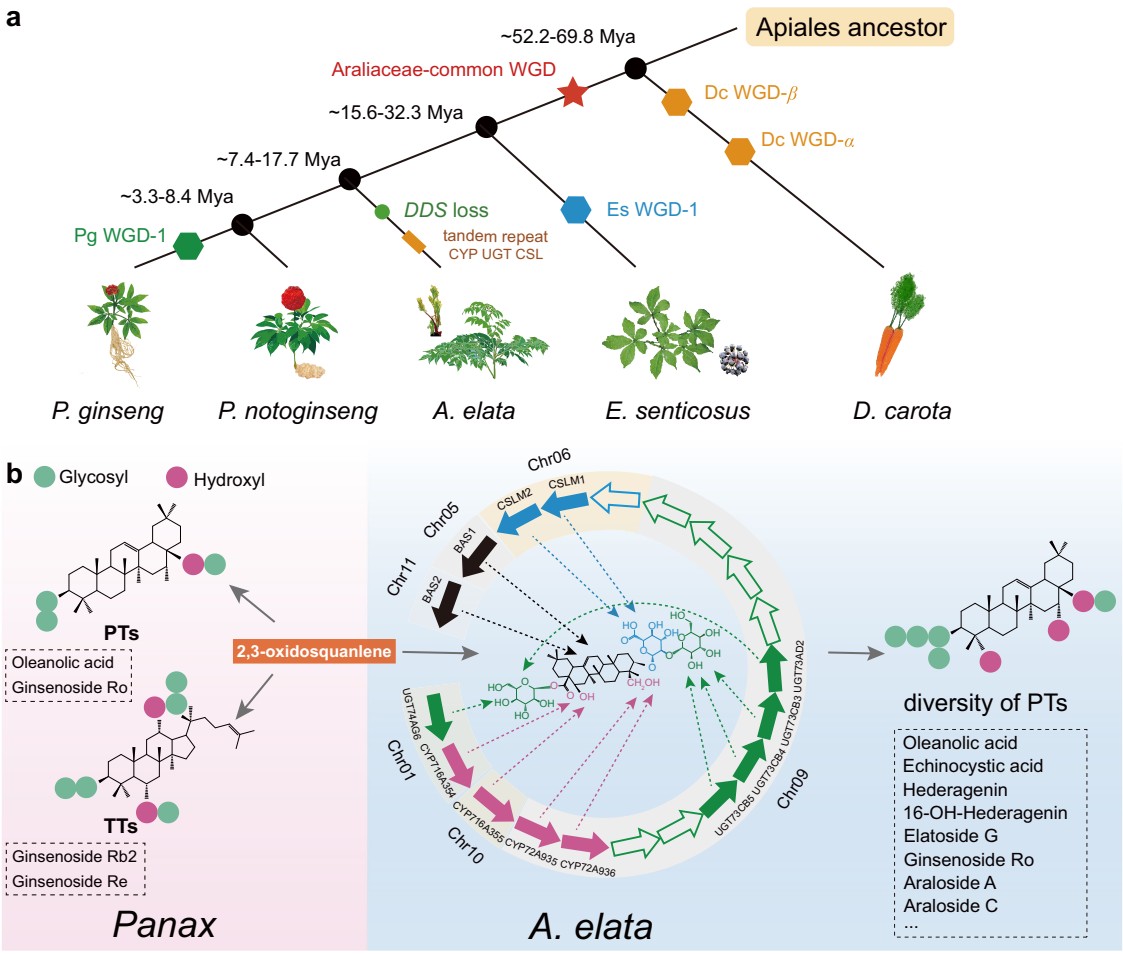

**Fig. 8 The species and metabolic evolution model of Araliaceae. a** Schematic diagram of divergence times in Apiales. The black points indicate the divergence-time nodes. **b** The pink and blue boxes represent the metabolites of *Panax* and *A. elata*, respectively. Genes involved in the alaroside biosynthetic pathway. The black arrow shows the cyclization of 2,3-oxidosqualene to β-amyrin catalyzed by BAS, the purple arrow shows the hydroxylation at the corresponding site catalyzed by P450, the blue arrow shows the glucuronosylation at the corresponding site by CSL, and the green arrows show the glycosylation reactions by the corresponding UGTs. The blank arrows indicate the pseudogenes in the tandem repeat that are not involved in the alaroside biosynthetic pathway. The compounds in dashed boxes represent the specific triterpenoids in *Panax* and *A. elata*. TTs tetracyclic triterpenoids, PTs pentacyclic triterpenoids.

all species. Source genes (core pPGs, core putative protogenes) were applied to look for colinearity blocks, and then we used anges software[69] to construct ancestral protochromosomes. Next, MCSCAN was used to find colinearity blocks for at least two species of orthologous genes (called pPG putative protogenes) but not including core pPGs, and then the colinearity blocks were mapped to the ancestral chromosome to construct ordered protogenes (oPG- ordered protogenes).

To analyze the whole-genome duplication events among *A. elata*, *D. carota*, and *V. vinifera* genomes, first, we found the best-homolog gene pairs in the species by BLASTp (-evalue 1e$^{-5}$ -max_target_seqs 20). Next, the $Ks$ analysis was estimated using the Nei–Gojobori method implemented in the YN00 program in the PAML (4.9 h)[70]. Finally, we integrated the $Ks$ value and block information by using WGDI (v0.5.1)[71].

**Transcriptome analysis**. Total RNA was extracted from four tissues (root, stem, leaf, and seed, with three biological repeats) using an RNA extraction kit (Huayueyang, ZH120, China). The mRNA from all samples was purified using the TruSeq RNA Sample Prep Kit (Illumina), followed by sequencing on an Illumina HiSeq 4000 sequencer. Raw RNA reads were filtered and trimmed to yield clean reads by Trimmomatic (v0.39)[72] and FastQC (v0.11.9)[73], and then these high-quality reads were mapped to the draft reference genome by HISAT2 with the following parameters: -dta-cufflinks. The gene expression values (FPKM) were calculated by FeatureCounts (v1.6.3)[74] with the following parameters: -p -C -B -P -O -T 16 -Q 20 -g gene_id -t exon. DESeq2[75] was used to normalize gene expression (BaseMean) in each sample, and differentially expressed genes (DEGs) were identified for each pair of compared groups by using P adj (adjusted p-value) < 0.05 as the threshold. The gene heatmap figures were produced by using TBtools with Heatmap Illustrator function.

**Chemicals**. Authentic standards of oleanolic acid, zingibroside R1, chikusetsusaponin IVa, and ginsenoside Ro were purchased from Shanghai Yuanye Bio-Technology Development Co.; hederagenin, echinocystic acid, 16-OH-hederagenin, oleanolic acid 28-O-glucopyranosyl ester, araloside C, and araloside A were purchased from Chengdu Purechemical-Standard Bio-Technology Development Co.; and calenduloside E was purchased from Chengdu Push Bio-Technology Development Co., and ginsenoside Re, Rf, Rg2, Rg1, F1, Rh1, Rc, Rb2, Rb1, Rd, and F2 were purchased from Shanghai Winherb Medical Technology Co. The internal standard digoxin was purchased from Sigma.

**Triterpene extraction and analysis**. Two-year-old *P. ginseng* plants and two-year-old *P. notoginseng* plants were collected from Jixi City, Heilongjiang Province, China, and Wenshan City, Yunnan Province, China, respectively. To detect the skeletons of triterpenoids in the roots, stems, and leaves of *P. ginseng*, *P. notoginseng*, and five-year-old *A. elata*, 20-mg aliquots were soaked in 2 mL of saponification solution [10% (w/v) KOH and 90% (v/v) ethanol]. Then, the samples were heated at 70 °C for 1 h, and the ethanol of the supernatant was evaporated. Next, 500 µL of ethyl acetate was added to dissolve the dried sample, 500 µL of ddH$_2$O was added, and the sample was centrifuged briefly after full vortexing. Then, 100 µL of the ethyl acetate layer was transferred to a new tube, and the samples were dried and methylated with a (trimethylsilyl) imidazole–pyridine mixture (Sigma, Shanghai, China). The methylated samples were analyzed by gas chromatography–mass spectrometry (GC–MS) using a Thermo TRACE 1310/ISQ LT gas chromatograph equipped with a TG-5HT (30 m × 0.25 mm × 0.10 µm) column (Thermo Fisher Scientific, USA). The GC conditions were as follows: the sample (1 µL) was injected into splitless mode at 250 °C under a He flow rate of 1.2 mL min$^{-1}$, the GC oven temperature was programmed to rise from an initial 170 °C for 2 min to 290 °C at

6 °C min$^{-1}$ and held at 290 °C for 4 min, after which the temperature was increased to 340 °C at 25 °C min$^{-1}$. The ion-trap heating temperature was 250 °C. Spectra were recorded between 15 and 28 min in the range of 60–800 $m/z$.

To detect the triterpenoids in all tissues of *P. ginseng*, *P. notoginseng*, and *A. elata*, we mixed the samples of roots, stems, and leaves equally. The 44-mg aliquots were extracted in 1 mL of 80% methanol containing 10 μg/mL digoxin as an internal standard. Samples were heated at 42 °C for 1 h and centrifuged for 5 min at 10,000 $g$, and the supernatant was filtered through a 0.22-μm membrane filter and double-diluted with 80% methanol for further analysis. Samples were analyzed by high-performance liquid chromatography–ultraviolet–mass spectrometry (HPLC–UV–MS/MS) using an AB Sciex TripleTOF 6600 LC/MS/MS instrument equipped with an ExionLC UHPLC unit (AB SCIEX, USA). Samples of 5 μL were injected onto a Kinetex C18 100 A column (Phenomenex, 150 × 4.6 mm, 2.6 μm). The column temperature was held at 40 °C, and the flow rate was held constant at 0.6 mL min$^{-1}$. A gradient using 0.1% v/v formic acid–water as Buffer A and 0.1% v/v formic acid–acetonitrile as Buffer B was run as follows: 5% Buffer B from 0 to 1 min, 5–95% Buffer B from 1 to 10 min, 95% Buffer B from 10 to 12 min, 95–100% Buffer B from 12 to 13 min, 100% Buffer B from 13 to 15 min, 100–5% Buffer B from 15 to 15.1 min, and 5% Buffer B until 18 min. A PDA detector was used to collect wavelengths from 190 to 700 nm. For mass spectrometry, the ionization mode was by electrospray ionization (ESI), the acquisition mode was information-dependent acquisition (IDA), and negative ion mode was used. The MS parameters were curtain gas, 35 psi; ion spray voltage, −4500 V; gas temperature, 550 °C; ion source gas 1 and gas 2, 55 psi; collision energy, 25 V; mass range, 50–1500 $m/z$; and acquisition rate, 20 frames s$^{-1}$. Chromatograms and mass spectra were analyzed using AnalystTF Software 1.7 (Sciex). Data were processed with PeakView 2.2 software (Sciex) and MasterView 1.3 software (Sciex). All analyses were performed in triplicate for each sample.

**Genetic transformation of A. elata cells and triterpenoid analysis.** The *A. elata* callus line was originally developed from the leaves of *A. elata* and was stably maintained in vitro on modified MS media supplemented with 3.0% sucrose, 0.01% myoinositol, 0.2 mg L$^{-1}$ 2,4-dichlorophenoxyacetic acid (2,4-D), 50 mM lipoic acid, and 200 μM acetosyringone. The cDNA of the *PgDDS* gene was cloned from *P. ginseng* roots and cloned into the PRI201-AN vector (TaKaRa, Shiga, Japan), generating plasmid PRI201-AN-*PgDDS*. *PgDDS* was transiently expressed in *P. ginseng* callus cells using *Agrobacterium rhizogenes* GV3101 (pSoup) (Weidi Biotechnology, Shanghai, China). A bacterial absorbance at A600 nm of 0.6 was used to infiltrate the *Agrobacterium* strain. The cells were cultured with *Agrobacterium* medium containing 200 μM acetosyringone in the dark for 2 days. Then, the cells were washed five times with water, transferred to selective MS agar medium containing 30.0 mg L$^{-1}$ kanamycin and 200 mg L$^{-1}$ timentin, and sub-cultured for two weeks in the dark at approximately 28 °C. Fresh callus cells (200 mg) were collected, and triterpenes were extracted using saponification solution as described above. The samples were dried and resolved using 200 μL of 80% methanol and analyzed by HPLC–MS as described above.

**Genome mining and phylogenetic analysis.** *OSC*, *P450*, *UGT*, and *CSL* genes were identified using HMMER3[76] with HMMER profiles (pHMMs) PF13243 and PF13249, PF00067, PF00201, and PF03552 downloaded from the PFAM library[77], respectively. Pseudogenes for OSCs, P450s, UGTs and CSLs (predicted protein sequence <300 amino acids) were discarded from the analysis. Retrieved sequences were aligned with MUSCLE, and phylogenetic trees were constructed using the maximum likelihood method with the JTT model and 1000 bootstrap replications in MEGA X software[78]. All the genes used for evolutionary analysis were listed in Supplementary Data 1.

**In vivo assays of the candidate OSC and CYP genes in A. elata.** The yeast strain WOE overexpressed four key enzymes, namely, the truncated HMG-CoA reductase gene (*tHMG1*), farnesyl pyrophosphate synthase gene (*ERG20*), squalene synthase gene (*ERG9*), and squalene epoxidase gene (*ERG1*) of the MVA pathway in WAT21: Mate α (*leu2–3,112 trp1–1 can1–100 ura3–1 ade2–1 his3–11 P$_{Gal1}$: ATR2–1*) was used for heterologous expression of OSC and P450 genes[79]. Two candidate *OSC* and 7 *P450* genes were cloned from cDNAs of *A. elata*. The coding sequence of AeS3 was synthesized by Genscript Corporation (Nanjing, China). The three *AeBAS*s were inserted into the BglII digestion sites in the pESC-TRP vector. The seven *P450* genes were inserted into the BglII digestion sites or SacII and XhoI digestion sites in the pESC-URA and pESC-HIS vectors to generate plasmids for heterologous expression in yeast (list in Supplementary Data 2). Next, the generated vectors were transformed into yeast strain WOE, respectively. The transformants were selected on synthetic defined (SD) medium without tryptophan, leucine, uracil, and histidine (-Trp, -Leu, -Ura, and -His) for new strains. The strains constructed in this study are summarized in Supplementary Data 3. Each colony was grown in 50-mL SD medium containing 2% glucose and precultured at 28 °C for 24 h at 200 rpm. Yeast cells were pelleted by centrifugation, rinsed with water, resuspended in 50 mL SD medium containing 2% galactose and 0.25 g L$^{-1}$ methionine for induction, and incubated at 28 °C for 5 days at 200 rpm. Yeast metabolites were extracted twice with 5 mL of *n*-butanol. The organic phases were dried and resolved in 200 μL of 80% methanol for further analysis by a modified

HPLC–MS method. The modified process was as follows: 5% Buffer B from 0 to 1 min, 5–95% Buffer B from 1 to 10 min, 95% Buffer B from 10 to 12 min, 95–5% Buffer B from 12 to 12.1 min, and then 5% Buffer B until 15 min.

**In vitro assays of the candidate UGT genes in A. elata.** The coding regions of *UGT* were cloned from cDNAs of *A. elata* and inserted into the BamHI and EcoRI digestion sites in the pGEX-4T-1 vector[80]. The primers are listed in Supplementary Data 4, and the plasmids are listed in the table of Supplementary Data 2. The recombinant pGEX-4T-1-*UGT* plasmids were then inserted into *E. coli* BL21 (DE3) for expression. The strains harboring the respective plasmids were grown in Luria-Bertani (LB) medium containing 100 μg/mL ampicillin at 37 °C. When the OD$_{600}$ reached 0.6–0.8, transcription in the cells was induced by incubation with 0.5 mM isopropyl-$\beta$-D-thiogalactopyranoside (IPTG) at 16 °C and 160 rpm for 20 h. The cell pellets were harvested by centrifugation at 4000 × $g$ for 10 min at 4 °C and then suspended in lysis buffer containing 50 mM Tris-HCl (pH 7.0). After sonication, the cell debris was centrifuged at 14,000 × $g$ and 4 °C. The supernatant was used as crude enzyme for enzymatic assays. The 500 μL enzymatic activity reaction mixture included 50 mM Tris-HCl (pH 7.0), 10 mM MgCl$_2$, 14 mM 2-mercaptoethanol, 0.1 mM acceptor substrate, 1 mM UDP-Glc, and 450 μL UGT crude protein. The reaction was incubated at 30 °C for 12 h and terminated by adding 500 μL of *n*-butanol. The residue was dissolved in 200 μL of 80% methanol and analyzed by a modified HPLC–UV–MS/MS method as described above.

**Purification and kinetic assay of UGTs.** The GST-tagged AeUGT73AD2, AeUGT74CB3, and AeUGT74AG6 specifically binded to glutathione immobilized to a matrix and can be easily separated from 200 mL cell lysate by a bind–wash–elute procedure of AKTA purifier (GE Healthcare) using a 1 mL GSTrap FF (GE Healthcare). For the kinetic assay of UGT using oleanolic acid, calenduloside E, zingibroside R1, and chikusetsusaponin IVa as substrates, the reaction conditions were tested using a 100-μL reaction mixture containing 50 mM Tris-HCl (pH 7.0), 1–300 μM substrates, 0.5 mM UDP-Glc, and 1 μM purified proteins at 30 °C for 1 h. Methanol (100 μL) was added to the reaction for HPLC–MS analysis.

**Cloning and heterologous expression of AeCSLM1 and AeCSLM2.** The coding regions of *AeCSLM1* and *AeCSLM2* were cloned into the pESC-LEU-EcUGD1 yeast galactose-induced expression vector using the primers listed in Supplementary Data 4. WAT21: Mate α (*leu2–3,112 trp1–1 can1–100 ura3–1 ade2–1 his3–11,15 P$_{Gal1}$: ATR2–1*) was transformed with the plasmid pESC-LEU-EcUGD1-AeCSLM1 and pESC-LEU-EcUGD1-AeCSLM2 to generate strains Y-CSLM1 and Y-CSLM2. WAT21 transformants were selected on an SD (-Leu) plate. Each colony was added to a 50-mL SD medium containing 2% glucose and cultured for 16 h at 28 °C. Yeast cells were then rinsed with water, resuspended in 50-mL SD medium containing 2% galactose, and cultured for 24 h at 28 °C. Each yeast strain was divided into 10-mL medium in each tube, 0.01 mM acceptor substrate (oleanolic acid and hederagenin, echinocystic acid, and 16-OH-heder-agenin) was added, and the cells were cultured for another 24 h at 28 °C for reaction. Yeast cells were harvested by 1.2 M sorbitol buffer and dialyzed (dissolved in 0.1 M sodium phosphate, pH 7.4) with 100 U lyticase for 2 h at 30 °C. Then, 500 μL of n-butanol was added to the mixture, and the new mixture was fully vortexed. The *n*-butanol layer was dried and resolved using 200 μL of 80% methanol and analyzed by the modified HPLC–MS method as described above.

**Strain construction for de novo synthesis of alarosides in yeast.** For the production of alarosides, *AeUGT74AG6* and *AeUGT73CB3* were inserted into the pESC-TRP-AeBAS1 and pESC-URA-AeCYP716A354 vectors, respectively. All vectors were inserted into strain WOE[79], which was favorable for triterpenoid synthesis and overexpressing four key genes of the MVA pathway. All plasmids are summarized in the table of Supplementary Data 2, and the primers are summarized in Supplementary Data 4. *S. cerevisiae* strains were transformed using the standard lithium acetate method[80]. All the strains used and generated in this study are summarized in the table of Supplementary Data 3. The transformants were selected on SD medium without tryptophan, leucine, uracil, and histidine (-Trp, -Leu, -Ura, and -His) for new strains. Each colony was grown in 50-mL SD medium containing 2% glucose and precultured at 28 °C for 24 h at 200 rpm. Yeast cells were pelleted by centrifugation, rinsed with water, resuspended in 50-mL SD medium containing 2% galactose and 0.25 g L$^{-1}$ methionine for induction, and incubated at 28 °C for 5 days at 200 rpm. Yeast metabolites were extracted twice with 5 mL of *n*-butanol. The organic phases were dried and resolved in 200 μL of 80% methanol for further analysis by the modified HPLC–MS method as described above.

**Structural analysis of the products.** Compound 5 was produced from a 10-L culture of yeast strain W8 (Supplementary Data 3) grown and induced by galactose. The yeast products were purified using a Waters 2487 HPLC (Waters, USA) coupled with an XB-C18 column (21.2 mm × 250 mm, 5-μm particle, Welch, Shanghai, China). The flow rate was held constant at 1 mL min$^{-1}$. A gradient using water as Buffer A and acetonitrile as Buffer B was run as follows: 70% Buffer B from 0 to 10 min, 70–100% Buffer B from 10 to 25 min, and 100–70% Buffer B from 25 to 30 min. A UV detector was used to collect wavelengths at 203 nm.

The spectra from 1D nuclear magnetic resonance (NMR, [1]H NMR and [13]C NMR) of compound 5 (dissolved in DMSO) were recorded with a AscendTM 800MHz spectrometer (Bruker, Karlsruhe, Germany). The chemical shifts were reported in parts per million (ppm) and referenced to the solvent.

**Reporting summary**. Further information on research design is available in the Nature Research Reporting Summary linked to this article.

## Data availability

The *A. elata* genome project has been deposited at the National Genomics Data Center under BioProject PRJCA006215. RNA-seq data have also been deposited at NCBI under BioProject no. PRJNA755350. The *A. elata* genome sequence, annotation information, gene expression abundance of different tissues, and paralogous gene information also deposited at Dryad Digital Repository [https://doi.org/10.5061/dryad.69p8cz937][81]. Nucleotide sequences of the genes reported in this work have been deposited in NCBI (Supplementary Table 13). HMM files from Pfam were used to predict CYP450 (PF00067), OSC (PF13249 and PF13243), UDPGT (PF00201), and CSL (PF03552) gene families. The *P. notoginseng* and *P. ginseng* transcriptome data involved in this study were obtained from the SRA database under accession SRX2253710 to SRX2253713 and Ginseng Genome Database [http://ginsengdb.snu.ac.kr/download.php?filename=Leaf_1Yr.tar.gz; http://ginsengdb.snu.ac.kr/download.php?filename=Root_6Yr.tar.gz; and http://ginsengdb.snu.ac.kr/download.php?filename=shoot.tar.gz], respectively. Source data are provided with this paper.

## Code availability

The in-house analysis scripts have been deposited in Github [https://github.com/Zeyu-An/A.elata-genome-NEFU] and Zenodo [https://doi.org/10.5281/zenodo.6373397], including scripts used for identification and visualization of gene families[82].

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

## Acknowledgements

This work was supported by the Innovation Project of State Key Laboratory of Tree Genetics and Breeding (Northeast Forestry University) (No. 2013A06) [Y.L.], the National Nonprofit Institute Research Grant of the Chinese Academy of Forestry (No. CAFYBB2019ZY003) [H.S.], National Natural Science Foundation of China (No. U21A20243) [Y.L.], the China Postdoctoral Science Foundation (No. 2015M581414) [Y.W.], and Heilongjiang Touyan Innovation Team Program (Tree Genetics and Breeding Innovation Team) [Y.L.]. The authors acknowledge the technical support by Dr. Shengnan Tan from Analysis and Test Center, Northeast Forestry University for assistance in LC–MS analysis and Dr. Xiaogang Niu at the mass spectrometry facility of the National Center for Protein Sciences at Peking University for assistance with NMR analysis. We also thank Prof. Chaosheng Yang of Yunnan Agricultural University for providing the genome sequence of *P. notoginseng*. We also thank Dr. Michael Court (Washington State University) of UGT nomenclature committee for the naming of the UGTs and Dr. David Nelson (The University of Tennessee Health Science Center) of P450 nomenclature committee for the naming of the P450s.

## Author contributions

Y.L., H.S., Z.-J.L., Z.P., and W.Y. planned and designed the research. H.Z., H.C.R., X.W., J.-N.Z., and D.R.Z. carried out experiments. Y.W., Z.A., D.-K.L., and D.Y.Z. analyzed the data. Y.W., H.Z., Z.A., P.W., H.W., J.Y., Z.Y.X., Z.C.X., Z.-J.L., and Y.L. interpreted the data and participated in discussion. Y.W., H.Z., and Z.A. wrote the paper. Y.L., H.S., Z.-J.L., Z.P., W.-C.T., Z.Y.X., and Z.C.X. revised the paper. The authors Y.W., H.Z., H.C.R., and Z.A. contributed equally to this work.

## Competing interests

The authors declare no competing interests.
