## [Peer Review File · Nature Communications]

Deletion and tandem duplications of biosynthetic genes drive the diversity of triterpenoids in *Aralia elata*Reviewers' Comments:

Reviewer #1:

Remarks to the Author:

This paper is a continuation of the evolving story about triterpenoid saponins in *Panax* species. It contains a vast amount of bioinformatics analysis and some biochemistry.

I am not an expert on the bioinformatics but interested in evolution of plant specialized metabolism, and my impression is that the paper would be much improved if the story was presented based on a few clear take home message that is backed up by a reasonable amount of selected data. There are 32 supplementary figures and 18 supplementary tables. In general the bioinformatics part could also be presented in a more clear way to the general reader of nature communication, as it is written now its more aimed for specialists. To increase the readability the legend of figure 1 could be revised, in particular figure 1a.

There is a nomenclature and a nomenclature committee for both P450s and UGTs. All UGTs and P450s should be named prior to publication to keep the value of the nomenclature systems intact.

ugt nomenclature:

<https://prime.vetmed.wsu.edu/resources/udp-glucuronosyltransferase-homepage/committee-members>

and

<https://prime.vetmed.wsu.edu/resources/udp-glucuronosyltransferase-homepage/current-nomenclature>

P450 nomenclature committee. Please send sequences to Dr Nelson. dnelson@uthsc.edu and <https://drnelson.uthsc.edu/>.

The recommend nomenclature is P450 and not CYP450. Also there is a misunderstanding of the concept of family and subfamily on line 236.

On line 250 the authors claim that the CYP72 subfamily member (AE09G02158) does not oxidize triterpenoids. But what are the positive controls for that the P450 is expressed in correctly folded and catalytically active state and not in an inactive form? Does it oxidize other substrates or can it produce a CO spectrum?

In eg figure 4, the stereo chemistry is shown. The data provided is LCMS, so how was the stereo chemistry concluded? Perhaps the authors be inspired by "Proposed minimum reporting standards for chemical analysis Chemical Analysis Working Group (CAWG) Metabolomics Standards Initiative (MSI) " <https://pubmed.ncbi.nlm.nih.gov/24039616/>

Inline 342; the reductase ATR1 does not provide NADPH to the P450, but it provides reducing equivalents from NADPH to the P450.

In supplementary figure 21 one sequenced is trimmed to the start codon and the other not – why? Same goes for supplementary figure 23 and the CSLMs.

The story about reconstruction of the pathway in callus show that post DDS genes are present, but why were plants not regenerated.

In supplementary data 28 the UGTs are called UDPGT and the in the text UGT. UGT would be according to nomenclature.

I am concerned about the UGT biochemistry and would like to see the authors compare their data to

data in the literature. How does the K_m compare with other known UGTs? A K_m of 32 mM does normally not indicate an in planta function. I would expect K_m s in the micro molar range and not millimolar range for an in planta function. Also, how does incubating with 5 mM UDPG for 12 hours compared to the literature? I would expect to see UDP glucose used at around 500 micro molar concentration and not 5 millimolar range, also doing enzyme kinetics over 12 hours does not seem optimal. There are report in the literature that plants UGTs normally have a K_m for UDP sugars around 250 micro molar. Finally, it seems like the same UGT names are not used in the table and main text, which is confusing.

Reviewer #2:

Remarks to the Author:

The manuscript by Li et al. presented a high-quality genome of *A. elata* to investigate the evolutionary mechanism of the compositional variations of dammarane-type and oleanane-type triterpenoid saponins between the *Panax* and *Aralia* genera, and determined that the functional loss of OSCs (DDS) and tandem duplication events of CSLs, CYP450s, and UGTs are crucial for the biosynthesis and accumulation of diverse aralosides in *A. elata*. In addition, they introduced the biosynthetic genes into yeast strains to produce various aralosides. This study provides a valuable information for isolating functional triterpenoid synthases and clarifying the evolutionary mechanism of triterpenoid biosynthesis by genomics and synthetic biology approaches, and will benefit the readers with similar background. However, some questions should be addressed before further consideration.

1. In order to help readers understand the compositional variations of dammarane-type and oleanane-type triterpenoid saponins between the *Panax* and *Aralia* genera, I suggest the authors include a table that provide these triterpenoid saponins from representative plants of *Panax* and *Aralia* genera.

2. In the Fig.3 d, the authors overexpressed the PgDDS gene in *A. elata* callus cells, and LC-MS analysis results showed that the overexpression of the PgDDS gene reconstructed the accumulation of dammarenediol and its derivative of protopanaxtriol in the transgenic lines of *A. elata*. What are the P450s and UGTs enzymes that catalyze these reactions in *A. elata*? The identification of derivatives and corresponding enzymes in the transgenic lines of *A. elata* is very important, because transferring the PgDDS gene to many other species may also produce dammarenediol, such as yeast, tobacco etc.

3. Besides dammaranediol synthase (DDS) in *P. ginseng* and *P. notoginseng*, DDS have been cloned from other plants, such as *Centella asiatica* (Apiaceae) (Plant Physiology and Biochemistry 47 (2009) 998–1002.), can this information be used for analysis?

4. Previous research showed that the biosynthetic pathways of dammarane-type triterpenes have been fully elucidated in *Panax* genera. For the quality of analysis the collinearity relationship of CYP72A, CSLM, and two UGT73 tandem duplication regions between *A. elata* and *P. notoginseng* (Fig.5), it will be important to provide data of the corresponding enzyme activity of *P. notoginseng* gene.

5. Part of synthetic biology and metabolic engineering

5.1 Some strains should be provided with more information, such as why they integrate the genes of ERG20, ERG9, ERG1, ATR2, PMet3-ERG7 etc.

5.2 Yeast strain W14-16, we can't find the description of plasmids pESC-URA-AECYP716A1-AEUGT74AG6, pESC-URA-AECYP716A1-AeUGT73CB3 and pESC-URA-AECYP716A1-AeUGT74AG6.

5.3 The authors engineered yeast for oleanane-type pentacyclic triterpenes production, what was the yield of products in yeast?

6. Part of Figures

6.1 As general rule graphics need to have also the reference units of Y axis, in this case the intensities.

6.2 Needs to provide the biological repeats of the FPKM values of gene expression in the roots, stems, and leaves of *A. elata*.

6.3 Figure 4a : "AeCYP72A1/AeCYP72A1" should be "AeCYP72A1/AeCYP72A2" ?

6.4 Figure 6a : It is useful to show the MS spectra of the authentic standards for comparison. If NMR

analysis is not available for products, at least high resolution MS characterization should be conducted.

7. Part of methods

7.1 Needs to provide the source information of WAT21, GV3101, and plasmids etc.

7.2 The ion spray voltage: " 4500 V" should be " - 4500 V"?

Reviewer #3:

Remarks to the Author:

This is a well written, well executed study to understand the biosynthesis of triterpene saponins in *Aralia elata* and how evolution impacted saponin biosynthesis in the the Araliaceae, specifically how *Aralia* diverged from *Panax* species in their triterpene profiles. The authors main discovery, that loss of exons in a key gene (dammaranediol synthase) in *Aralia* lead to the inability to produce dammarane-type saponins in *Aralia*. The authors then did a comprehensive survey of tailoring enzymes in *A. elata* and then performed a wide range of functional studies in *A. elata* and in yeast to demonstrate gene function of these tailoring enzymes. From this work, the authors generated not only a nice model of evolution of saponin biosynthesis in the Araliaceae, but also a knowledge bank of the biosynthetic pathway.

The genomics and bioinformatics methods are solid and well described, including appropriate supplemental files to support their interpretations. Specific suggestions on the manuscript are:

Line 111: replace repetition with repetitive sequences

Line 400: describe the rounds of error correction with Pion

Add an analysis of heterozygosity of the final genome. It seemed quite heterozygous in the k-mer analysis and it would be good to know how well the consensus assembly represents the two haplotypes.

While the sequences have been put in the NCDC, I think it would be extremely helpful for the community to have a Dryad Digital Repository with not only the genome sequence, annotation but also the gene expression abundances, orthologous groups, etc.

Either provide the code via GitHub (or its equivalent) or remove the statement on code availability. It is impractical to request code from authors.

Add a figure in the Introduction on the basic biochemistry of triterpene saponins and how *Panax* and *Aralia* differ; this would engage non-terpene readers.

Most figures are too small to read, consider reframing these so that all the features can be seen- especially Fig 1b, 2c, all of Fig 5, 6b, all of Fig 7, 8c

Supp Fig 13 is quite informative. This would merit moving to the main manuscript.

Reviewer #4:

Remarks to the Author:

This manuscript proposes chromosome level genome sequences of *Aralia elata* (Ae) which is the closest genus of *Panax* species and unveil the biosynthetic pathways and genes for diversity of triterpene biosynthesis. The manuscript supports the plant genome evolution of Araliaceae family and report the causal genes for loss of dammarane-type ginsenoside in *Aralia* genus. They found loss of 12

exons in dammarediol synthase-encoding gene in *Ae* and confirmed the complementation test by recovery of dammarediol-type saponins mediated by overexpression of normal *Panax ginseng* DDS gene (PgDDS) in *Ae* callus. They also found that tandem duplication events of triterpene biosynthetic genes function for biosynthesis of oleanane-type ginsenoside and aralosides. They also synthesized 13 aralosides *in vivo* using *Saccharomyces* system.

Overall, the manuscript is well written and show the very unique biosynthesis pathway based on the finding of causal genes using newly assembled high-quality *Ae* genome sequence. I have a few issues to clarify for this manuscript because the paper can be a golden standard for the genomics research in the Araliaceae family which by far has no clear reference genome sequence.

1. The *Ae* plant might have high heterozygosity. The overall genome assembly might be correct. However, the pseudochromosome sequence is not supported by genetic map or high-resolution FISH analysis using chromosome specific oligomers. Super-scaffolding by Hi-C analysis can induce large scale genome mis-assembly. If some efforts are taken to evaluate the pseudochromosome sequence, the *Ae* genome sequence can be widely utilized as the reference for comparative study with other Araliaceae genomes.

2. I have concern about Dc-beta genome duplication which was represented in Apiaceae unique genome triplication [or sometime as duplication in the *Eleutherococcus senticosus* genome paper (Yang et al. 2021. Molecular Ecology Resources)]. The *Daucus carota* genome paper reported two genome duplications, Dc-beta (triplication) and Dc-alpha (duplication) (Iorizzo et al. 2016. Nature Genetics). However, when the *Panax ginseng* (Pg) genome assembly were compared to the *D. carota* genome, they found only two syntenic blocks against four Pg genomes which is derived from Pg-alpha and Pg-beta genome duplications (Kim et. al, 2018 Plant Biotechnology). The *Ae* genome assembly showed clear synteny for Pg-beta with *Vitis* genome which was diverged 120 million years ago. It is clear that Pg-beta genome duplication share with the *Ae* genome duplication. However, I wonder whether Dc-beta is really an Apiaceae unique event. In this manuscript, there are many genome level comparisons but lack genome comparison between *Ae* and Dc genomes. I'd like to ask for genome level comparisons to show the Dc-beta genome triplication event to clearly support figures 1 and 2. Some minor specific comments are listed below:

1. Line 211: Supplementary Fig 15 \diamond Supplementary Fig. 16

2. Line 217: "post-modification" to "post-transcriptional modification" or "post-translational modification"?

3. Line 218: spelling of "A. *ellata*"

4. Line 232: wrong figure citation. Fig. 4b \diamond 4c

5. Line 244 vs 250: CYP72A and CYP72, are they the same? Please be consistent with italics for genes, and please clarify the difference between suffixes like -A.

6. Fig 3a: Emphasize the two genes (AE11G00256.1 and AE05G00216.1) by making them bold in the phylogenetic tree for easy access.

7. Fig. 4b: You made some CYP716A genes bold to emphasize but the rest (AE11G00717.1 and AE09G02158.1) were not. Any special reason? Please emphasize all for easy visual access.

8. Fig. 4c: The yeast strains with the AeBAS1+AeCYP716A1+AeCYP716A2+AeCYP72A2 showed 16-OH-hederagenin peak in the chromatogram, which contradicts the pathway in 4a and your claim in lines 250-252. Based on Fig. 4a, the AeBAS1+AeCYP716A1+AeCYP716A2+AeCYP72A2 construct should only accumulate Echinocystic acid and no 16-OH-hederagenin as CYP72 is unable to process upstream compounds. Is your diagram in Fig. 4a wrong (the names of enzymes, eg. AeCYP72A1/AeCYP72A1)?

9. Fig. 5a,c,e,g: There is no description of what the blue bar means in the legend.

10. Lines 268-270: Wrong figure reference. There is no genome evolution figure in Fig.5a

11. Line 280,286: Do you mean Fig. 5c,d?

12. Line 292: Wrong figure citation.

13. Line 295: There is no Fig. 4f. You mean 5f? The use of different nomenclature is confusing. Please be consistent with what gene names you use. What is AeUGT73CB3? I could not see this in Fig. 5, where you use different gene names, i.e. AE..1544.

14. Line 352: Supplementary Fig. 17: Did you mark one gene incorrectly? (AE04G0058.1 is not one of the two selected BAS genes).

“Point-by-Point” Responses to Reviewers’ Comments:

Manuscript Number: NCOMMS-21-37280

Manuscript title: Partial deletion and tandem duplications of biosynthetic genes drive the diversity of triterpene saponins in *Aralia elata*

Reviewer #1:

Comment 1: This paper is a continuation of the evolving story about triterpenoid saponins in *Panax* species. It contains a vast amount of bioinformatics analysis and some biochemistry. I am not an expert on the bioinformatics but interested in evolution of plant specialized metabolism, and my impression is that the paper would be much improved if the story was presented based on a few clear take home message that is backed up by a reasonable amount of selected data. There are 32 supplementary figures and 18 supplementary tables. In general the bioinformatics part could also be presented in a more clear way to the general reader of nature communication, as it is written now its more aimed for specialists. To increase the readability the legend of figure 1 could be revised, in particular figure 1a.

Response 1: We are grateful to the reviewer for taking the time to assess our manuscript and to provide insightful and constructive comments. We have now revised the manuscript based on the suggestions, which have improved our manuscript. To help the general reader of *Nature Communication* to understand our story, we added a new figure (Fig.1) in the introduction to show the structural and compositional differences in triterpenoid saponins among the selected *Panax* and *Aralia* species. We also revised the bioinformatics part of the current version to emphasize the important questions in genome evolution, particularly about the evolution of gene families and genome structure, which may result in compositional differences in the triterpenoid saponins between *A. elata* and *Panax* (please see lines 145-150, 161-167, 175-185). Moreover, the Figure 1a in previous version have been moved to Supplementary Figure 4 of the revised manuscript, and its legend have been modified to improve the readability.

Comment 2: There is a nomenclature and a nomenclature committee for both P450s and UGTs. All UGTs and P450s should be named prior to publication to keep the value of the nomenclature systems intact.

ugt nomenclature:

<https://prime.vetmed.wsu.edu/resources/udp-glucuronosyltransferase-homepage/committees-members>

and

<https://prime.vetmed.wsu.edu/resources/udp-glucuronosyltransferase-homepage/current-nomenclature>

P450 nomenclature committee. Please send sequences to Dr Nelson. dnelson@uthsc.edu and <https://drnelson.uthsc.edu/>.

Response 2: Thank you for your suggestions. All the UGTs and P450s have been named by Dr. Michael and Dr. Nelson in the current version.

Comment 3: The recommend nomenclature is P450 and not CYPP450. Also there is a misunderstanding of the concept of family and subfamily on line 236.

Response 3: We thank the reviewer for spotting this error. All the “CYPP450” have been changed to “P450”. We have corrected this error on line 252 of the revised manuscript.

Comment 4: On line 250 the authors claim that the CYP72 subfamily member (AE09G02158) does not oxidize triterpenoids. But what are the positive controls for that the P450 is expressed in correctly folded and catalytically active state and not in an inactive form? Does it oxidize other substrates or can it produce a CO spectrum?

Response 4: Thank you very much for this comment. We carefully redone the experiments for functional analysis of AE09G02158, and found that, AE09G02158 (AeCYP72A937) showed very weak activity to catalyze the hydroxylation at the C23 position of the hederagenin, leading to the formation of 16-OH-hederagenin (Fig 4c). We updated this information in the revised manuscript (please see page 11, lines 267-268).

Comment 5: In eg figure 4, the stereo chemistry is shown. The data provided is LCMS, so how was the stereo chemistry concluded? Perhaps the authors be inspired by “Proposed minimum reporting standards for chemical analysis Chemical Analysis Working Group (CAWG) Metabolomics Standards Initiative (MSI)

“ <https://pubmed.ncbi.nlm.nih.gov/24039616/>

Response 5: We have provided NMR data (Supplementary Figures 30), which we used to analyze the stereochemical structure of chemicals in Figure 4.

Comment 6: Inline 342; the reductase ATR1 does not provide NADPH to the P450, but it provides reducing equivalents from NADPH to the P450.

Response 6: Thank you. We have corrected that on line 368 of the revised manuscript.

Comment 7: In supplementary figure 21 one sequenced is trimmed to the start codon and the other not – why? Same goes for supplementary figure 23 and the CSLMs.

Response 7: Thank you for noticing this. We found that the corresponding ATG translational start codon of AE09G00613 and AE06G00234 are upstream of the annotated start codon in the *Aralia elata* genome; they were annotated based on sequence homology and *de novo* gene predictions. We have re-annotated these two genes and changed the Supplementary Figure 28 and 32 based on the new alignments.

Comment 8: The story about reconstruction of the pathway in callus show that post DDS genes are present, but why were plants not regenerated.

Response 8: We thank the reviewer for this concern. The callus cell from *Aralia elata* used in the DDS transformation experiment is non-embryogenic callus, which was stably propagated over 40 generations. The callus showed strong cell proliferation ability, and genetic transformation was efficient. If we induce the callus to switch from the “proliferation” phase to the “redifferentiation” phase, it will take over one year to regenerate transgenic plants. To restore DDS activity in *A. elata*, we used the callus in a gene complementation experiment and detected dammarenediol and protopanaxatriol in the transgenic cell lines that overexpressed the *PgDDS* gene. The results demonstrate that the loss of the *DDS* gene in the *A. elata* genome is the key factor for the lack of dammarane-type saponins in *A. elata*.

Comment 9: In supplementary data 28 the UGTs are called UDPGT and the in the text UGT. UGT would be according to nomenclature.

Response 9: Thank you for spotting this error; it has now been corrected in Supplementary Figure 38 of the revised manuscript.

Comment 10: I am concerned about the UGT biochemistry and would like to see the authors compare their data to data in the literature. How does the Km compare with other known UGTs? A Km of 32 mM does normally not indicate an in planta function. I would expect Kms in the micro molar range and not millimolar range for an in planta function. Also, how does incubating with 5 mM UDPG for 12 hours compared to the literature? I would expect to see UDP glucose used at around 500 micro molar concentration and not 5 millimolar range, also doing enzyme kinetics over 12 hours does not seem optimal. There are report in the literature that plants UGTs normally have a Km for UDP sugars around 250 micro molar. Finally, it seems like the same UGT names are not used in the table and main text, which is confusing.

Response 10: According to your suggestions, we did enzyme kinetics of UGTs with 0.5 mM UDPG for 1 hour. These UGTs have a Km for UDPG around 40-650 micro molar, close to the Km of UGTs from ginseng and *Siraitia grosvenorii*, which were reported before (Jung et al., *Plant Cell Physiol.* 2014, 55(12): 2177–2188; Li et al., *ACS Catal.* 2020, 10, 3629–3639). We have updated the results on lines 325-329 and lines 357-363. To be clearer, we changed all the “gene IDs” for AeUGTs to “gene names” in Figures 5 to 7 of the revised manuscript.

Reviewer #2 (Remarks to the Author):

The manuscript by Li et al. presented a high-quality genome of *A. elata* to investigate the evolutionary mechanism of the compositional variations of dammarane-type and oleanane-type triterpenoid saponins between the *Panax* and *Aralia* genera, and determined that the functional loss of OSCs (DDS) and tandem duplication events of CSLs, CYP450s, and UGTs are crucial for the biosynthesis and accumulation of diverse aralosides in *A. elata*. In addition, they introduced the biosynthetic genes into yeast strains to produce various aralosides. This study provides a valuable information for isolating functional triterpenoid synthases and clarifying the evolutionary mechanism of triterpenoid

biosynthesis by genomics and synthetic biology approaches, and will benefit the readers with similar background. However, some questions should be addressed before further consideration.

Response: We thank the reviewer for his/her careful and overall positive evaluation of our manuscript.

Comment 1: In order to help readers understand the compositional variations of dammarane-type and oleanane-type triterpenoid saponins between the *Panax* and *Aralia* genera, I suggest the authors include a table that provide these triterpenoid saponins from representative plants of *Panax* and *Aralia* genera.

Response 1: Thank you for this constructive concern. Inspired by reviewers, we have provided an illustration to show the structure and composition of different triterpenoid saponins from *Panax* and *Aralia* genera in Fig. 1 in the revised manuscript.

Comment 2: In the Fig.3 d, the authors overexpressed the PgDDS gene in *A. elata* callus cells, and LC-MS analysis results showed that the overexpression of the PgDDS gene reconstructed the accumulation of dammarenediol and its derivative of protopanaxatriol in the transgenic lines of *A. elata*. What are the P450s and UGTs enzymes that catalyze these reactions in *A. elata*? The identification of derivatives and corresponding enzymes in the transgenic lines of *A. elata* is very important, because transferring the PgDDS gene to many other species may also produce dammarenediol, such as yeast, tobacco etc.

Response 2: We thank the reviewer for this concern. We have identified the functional protopanaxadiol synthase in *A. elata*, AeCYP716A356, which was able to catalyze the formation of protopanaxadiol from dammarenediol II when it was expressed heterologously in yeast (Supplementary Figure 25). In addition, the overexpression of the PgDDS gene also led the accumulation of hydroxylated derivatives of dammarenediol, indicating that the tailoring enzymes for formation of dammarane-type saponins are functional and highly conserved between *Panax* and *Aralia* genera. The key reason that *A. elata* does not accumulate the dammarane-type saponins is the loss of 12 exons of the DDS gene during the evolution of the *A. elata* genome. We have included these results in the revised manuscript (please see page 10, lines 229-232).

Comment 3: Besides dammaranediol synthase (DDS) in *P. ginseng* and *P. notoginseng*, DDS have been cloned from other plants, such as *Centella asiatica* (Apiaceae) (Plant Physiology and Biochemistry 47 (2009) 998–1002.), can this information be used for analysis?

Response 3: We thank the reviewer for this information and question. Our phylogenetic analysis of all the dammaranediol synthases from *A. elata*, *P. ginseng*, *P. notoginseng* and previously reported DDS genes from other plants indicates that a DDS-like protein (AE10G00915) has 85% sequence similarity to CaDDS (*Centella asiatica*) (Supplementary Figure 19). But, AE10G00915 is not expressed in any tissue analyzed in the *A. elata*. Moreover, the orthologues of CaDDS in *P. ginseng* (PgS4166.7) and *P. notoginseng* (Pno10G001491.t1) are also not expressed in any tissue (Supplementary Figure 20). The protein coding region of the AE10G00915 was

synthesized and then expressed in the yeast strain WAT21. Only α -amyirin and β -amyirin were detected in the resulting strain, indicating AE10G00915 is a multifunctional OSC producing α - and β -amyirin in a heterologous yeast host, but not a dammaranediol II synthase (**Supplementary Figure 21**). Hence, we named AE10G00915 as AeAS3, and included these results in the revised manuscript (please see page 9, lines 207-214).

Comment 4: Previous research showed that the biosynthetic pathways of dammarane-type triterpenes have been fully elucidated in *Panax* genera. For the quality of analysis the collinearity relationship of CYP72A, CSLM, and two UGT73 tandem duplication regions between *A. elata* and *P. notoginseng*(Fig.5), it will be important to provide data of the corresponding enzyme activity of *P. notoginseng* gene.

Response 4: We have added the enzymatic assay data for the colinear enzymes in *P. notoginseng*, including Pno09G000253(CYP72A), Pno06G01541(CSLM), Pno14G001407(UGT73), and Pno14G001424(UGT73) in **Supplementary Figure 37**. These results showed that these colinear genes in *A. elata* and *P. notoginseng* have similar functions. We have included these results and descriptions in the revised manuscript (please see page 14, lines 338-343).

Comment 5: Part of synthetic biology and metabolic engineering

Comment 5.1: Some strains should be provided with more information, such as why they integrate the genes of ERG20, ERG9, ERG1, ATR2, PMet3-ERG7 etc.

Response 5.1: We have provided the information and details about the metabolic engineering of yeast strains on lines 634-637 and 689-692.

Comment 5.2: Yeast strain W14-16, we can't find the description of plasmids pESC-URA-AECYP716A1-AEUGT74AG6, pESC-URA-AECYP716A1-AeUGT73CB3 and pESC-URA-AECYP716A1-AeUGT74AG6.

Response 5.2: We have now provided the description of these plasmids in Supplementary table 13.

Comment 5.3: The authors engineered yeast for oleanane-type pentacyclic triterpenes production, what was the yield of products in yeast?

Response 5.3: The yield of oleanane-type pentacyclic triterpenes in yeast were quite low, about 1 $\mu\text{g/L}$ -20 $\mu\text{g/L}$, which may due to low catalytic capability of the plant-derived glycosyltransferase in yeast (Zhuang et al., *Metabolic Engineering*, 2017, 24: 25-32). More metabolic engineering strategies, such as directed evolution, will be designed in future work to increase the yield of oleanane-type pentacyclic triterpenes.

Comment 6: Part of Figures

Comment 6.1: As general rule graphics need to have also the reference units of Y axe, in this case the intensities.

Response 6.1: We have added the reference units for the y-axes and intensities in each figure in the revised manuscript.

Comment 6.2: Needs to provide the biological repeats of the FPKM values of gene expression in the roots, stems, and leaves of *A. elata*.

Response 6.2: Three biological replicates were used for RNA-seq analyses of different tissues from *A. elata*. We averaged the FPKM values from replicate samples for drawing the heatmaps in Fig. 4 and Supplementary Figure 39. These descriptions have been provided in the methods and figure legends.

Comment 6.3: Figure 4a: “AeCYP72A1/AeCYP72A1” should be “AeCYP72A1/AeCYP72A2”?

Response 6.3: We thank the reviewer for the good catch. It has now been corrected in Figure 4a.

Comment 6.4: Figure 6a: It is useful to show the MS spectra of the authentic standards for comparison. If NMR analysis is not available for products, at least high resolution MS characterization should be conducted.

Response 6.4: We thank the reviewer for this constructive suggestion. We have provided the MS spectra of the authentic standards for compounds 1 to 10 in the **Figure 6b** and **Supplementary Figures 42 and 43** in the revised manuscript. The standards of compounds 11 to 14 are not commercially available, and their low contents in the engineered yeast strains precluded their NMR spectroscopic analysis. Nevertheless, we deduced the molecular structures of these new compounds according to the following four lines of evidence.

1. The structures of compounds 11 to 14 are very similar to the glycosylated oleanolic acid derivatives 6 to 9 (only lacking of the -OH at the C16 position). These compounds were glycosylated by AeCSLM1, AeUGT74AG6, and AeUGT73CB3 from Oleanolic acid, and Echinocystic acid, respectively (Figure 6a).
2. The AeUGT74AG6 and AeUGT73CB3 characterized in this study displayed high regiospecificity to a specific position of the triterpene scaffolds, but no tight substrate specificity. For example, AeUGT74AG6 transferred a glucosyl moiety to the free C28-OH of Oleanolic acid, Calendulose E and Zingibroside R1 to produce Oleanolic acid 28-O-glucopyranosyl ester, Chikusetsusaponin Iva and Ginsenoside Ro, respectively (**Supplementary Figure 40**). AeUGT73CB3 catalyzed the second glucosylation of the hydroxyl group at the C3 position in Calendulose E and Chikusetsusaponin Iva to produce Zingibroside R1 and Ginsenoside Ro, respectively (**Supplementary Figure 35**).
3. The orthologs of AeCSLM1 in soybean and spinach, GmCSyGT3 and SoCslG, also showed substrate promiscuity. They catalyzed the C3 position-specific glucuronosylation of diverse oleanane-type triterpenoid substrates, such as oleanolic acid, glycyrrhetic acid, soyasapogenol B, augustic acid, hederagenin, gypsogenin and gypsogenic acid (Chung et al., Nature Communications. 2020, 11:5664; Jozwiak et al., Nature Chemical Biology. 2020, 16: 740-748). Moreover, all the reported glycosyltransferases from the cellulose synthase-like M (CslM) subfamily showed unequivocal specificity toward UDP-GlcA as the glycosyl donor.

4. *In vitro* catalytic experiments of these three glycosyltransferases with corresponding substrates combined with mass spectrometry analysis further confirmed the molecular weights of these compounds (**Supplementary Figures 33, 35, 40, and 43**).

We have included these results and descriptions in the revised manuscript (please see page 16, lines 387-391).

Comment 7: Part of methods

Comment 7.1: Needs to provide the source information of WAT21, GV3101, and plasmids etc.

Response 7.1: Done (in Method and Materials).

Comment 7.2: The ion spray voltage: “ 4500 V” should be” - 4500 V”?

Response 7.2: We thank the reviewer for spotting this error. It has now been corrected.

Reviewer #3 (Remarks to the Author):

This is a well written, well executed study to understand the biosynthesis of triterpene saponins in *Aralia elata* and how evolution impacted saponin biosynthesis in the the Araliaceae, specifically how *Aralia* diverged from *Panax* species in their triterpene profiles. The authors main discovery, that loss of exons in a key gene (dammaranediol synthase) in *Aralia* lead to the inability to produce dammarane-type saponins in *Aralia*. The authors then did a comprehensive survey of tailoring enzymes in *A. elata* and then performed a wide range of functional studies in *A. elata* and in yeast to demonstrate gene function of these tailoring enzymes. From this work, the authors generated not only a nice model of evolution of saponin biosynthesis in the Araliaceae, but also a knowledge bank of the biosynthetic pathway.

The genomics and bioinformatics methods are solid and well described, including appropriate supplemental files to support their interpretations.

Response: We are delighted that the reviewer found our studies thorough and expertly conducted, our data are of high significance. We also appreciate the helpful comments and suggestions.

Specific suggestions on the manuscript are:

Comment 1: Line 111: replace repetition with repetitive sequences.

Response 1: We thank the reviewer for spotting this error. We have now corrected this on page 5, line 110.

Comment 2: Line 400: describe the rounds of error correction with Pilon.

Response 2: We thank the reviewer for this comment. Two rounds of corrections have been made to the draft assembly from PacBio reads using Pilon. We have added this information to page 18, line 435 of the main text.

Comment 3: Add an analysis of heterozygosity of the final genome. It seemed quite

heterozygous in the k-mer analysis and it would be good to know how well the consensus assembly represents the two haplotypes.

Response 3: By mapping Illumina short reads back to the draft genome, we observed a heterozygosity rate of 0.57% in the final genome (Table R1). This rate is smaller than the estimate heterozygosity by *K*-mer analysis, which is 1.62%. The difference between these two rates is due to the removal of the redundant contigs during the assembly process (Figure R1). By using all the PacBio continuous long reads (CLR), we carried out a preliminary genome assembly with a total length of 1,747,086,597 bp, which is much larger than the genome size estimated by genome survey analysis (Table R2). However, the total length of the preliminary assembled sequence is less than two times the estimated genome size, which indicates the heterozygosity rate of preliminary genome assembly is not high enough for further assembling into two sets of contiguous pseudo-haplotypes. We compared the distribution of per-base sequencing depth between the preliminary and final genome using minimap2. The result showed that the single-base depth distribution of heterozygous rate was significantly reduced in the final genome (Figure R1). For this diploid genome, the final assembly genome is a mosaic of the two haplotypes. In addition, we found that the mapping rate of Illumina short reads to the final genome was 99.52%, indicating that the resulting assembly represents one artificial consensus sequence with two haplotypes merged. We added the heterozygosity analysis of the final genome on lines 121-123.

Table R1 Number of heterozygous sites in the final genome.

Chromosome number	Length (bp)	Sequencing depth ≥ 5 sites	Number of heterozygous sites	Percentage heterozygous sites
Chr01	94211576	93847012	407016	0.43
Chr02	96582347	95675787	584105	0.61
Chr03	73159195	72619159	554691	0.76
Chr04	88419863	87890504	567018	0.65
Chr05	52958921	52649833	441603	0.84
Chr06	108756462	107632357	521949	0.48
Chr07	85555957	84812361	512612	0.6
Chr08	83843920	83495614	414119	0.5
Chr09	110444823	108583460	549800	0.51
Chr10	79603388	79018015	360379	0.46
Chr11	83455037	82947016	547164	0.66
Chr12	81024418	80261360	491876	0.61
unchr_scaffold_1	1678530	1672438	893	0.05
unchr_scaffold_2	63875	61305	0	0.0
unchr_scaffold_3	177834	176166	2	0.0
unchr_scaffold_4	1320585	1316969	871	0.07
unchr_scaffold_5	1305450	1295979	1035	0.08
unchr_scaffold_6	88157	85628	1	0.0
unchr_scaffold_7	118715	117876	57	0.05

unchr_scaffold_8	408329	404726	73	0.02
unchr_scaffold_9	317950	316000	355	0.11
unchr_scaffold_10	2142366	2136897	9003	0.42
unchr_scaffold_11	1006328	1004219	340	0.03
unchr_scaffold_12	57337	56228	108	0.19
unchr_scaffold_13	95921	93964	46	0.05
unchr_scaffold_14	334581	331245	19	0.01
unchr_scaffold_15	49632	46262	0	0.0
Total	1047181497	1038548380	5965135	0.57

Figure R1 Distribution of single-base sequencing depth before and after removing the redundant contigs.

Table R2 Assembly statistics before and after removing the redundant contigs.

Statistics	Canu		Final	
	Size (bp)	Number	Size (bp)	Number
N90	103,362	2,911	245,166	936
N80	229,198	1,762	468,306	634
N70	389,106	1,176	706,971	454
N60	567,491	806	953,751	326
N50	772,368	542	1,195,804	229
Longest	17,272,275	-	17,272,275	-

Total Length	1,747,086,597	-	1,046,177,497	-
Total number (≥ 100 bp)	-	5,950	-	2,035
Total number (≥ 2000 bp)	-	5,950	-	2,035
GC rate	0.355	-	0.355	-

For easy reference, the analytical methods are mentioned here.

Genome mapping

We used BWA to index the *Aralia elata* genome with 'bwa index -a is'. Then we mapped Illumina short reads to the final genome using BWA-MEM with default options.

Heterozygosity analysis

We identified SNPs using GATK with 'gatk --java-options -Xmx4G HaplotypeCaller' and extracted SNP information with 'gatk SelectVariants -select-type-to-include SNP'. We filtered low quality SNP sites using 'gatk VariantFiltration --filter-expression 'QUAL < 30.0 || QD < 2.0 || FS > 60.0 || SOR > 4.0' --filter-name lowQualFilter --cluster-window-size 10 --cluster-size 3 --missing-values-evaluate-as-failing'. Finally, we calculated the heterozygosity of SNP site that had a sequencing depth greater than 5.

Single-base sequencing distribution analysis

We mapped the PacBio long reads to the preliminary genome assembled by canu and to the final genome using minimap2 with 'minimap2 -ax map-pb'. The sequencing depth of each site in the genome was then determined.

Comment 4: While the sequences have been put in the NCDC, I think it would be extremely helpful for the community to have a Dryad Digital Repository with not only the genome sequence, annotation but also the gene expression abundances, orthologous groups, etc.

Response 4: We have submitted the *A. elata* genome sequence, annotation information, gene expression abundance of different tissues, and paralogous genes to Dryad Digital Repository (reviewer access: <https://datadryad.org/stash/share/1WEQniWXq9fZQAtnBRAJczpkKpHqaBxp2k43YUgfYFA>; reader access: <https://doi.org/10.5061/dryad.69p8cz937>). DATA AVAILABILITY in the main text has also been updated with this information on lines 725-727.

Comment 5: Either provide the code via GitHub (or its equivalent) or remove the statement on code availability. It is impractical to request code from authors.

Response 5: We appreciate this advice. For analyzing homologous genes in this article, we used the JCVI pipeline (<https://github.com/tanghaibao/jcvi>). The analysis scripts have been deposited in Github (<https://github.com/Zeyu-An/A.elata-genome-NEFU>). This information has been updated in the revised manuscript on lines 729-731.

Comment 6: Add a figure in the Introduction on the basic biochemistry of triterpene saponins and how Panax and Aralia differ; this would engage non-terpene readers.

Response 6: As noted earlier, we have added an illustration to show the structure and composition of different triterpene saponins from *Panax* and *Aralia* in Figure 1.

Comment 7: Most figures are too small to read, consider reframing these so that all the features can be seen-especially Fig 1b, 2c, all of Fig 5, 6b, all of Fig 7, 8c.

Response 7: We modified all the figures and increased the font sizes where possible to improve readability.

Comment 8: Supp Fig 13 is quite informative. This would merit moving to the main manuscript.

Response 8: We changed the bar chart in Supplementary Figure 13 of the previous version to a pie chart and moved it to Figure 1 in the revised manuscript.

Reviewer #4 (Remarks to the Author):

This manuscript proposes chromosome level genome sequences of *Aralia elata* (Ae) which is the closest genus of *Panax* species and unveil the biosynthetic pathways and genes for diversity of triterpene biosynthesis. The manuscript supports the plant genome evolution of Araliaceae family and report the causal genes for loss of dammarane-type ginsenoside in *Aralia* genus. They found loss of 12 exons in dammarediol synthase-encoding gene in Ae and confirmed the complementation test by recovery of dammarediol-type saponins mediated by overexpression of normal *Panax ginseng* DDS gene (PgDDS) in Ae callus. They also found that tandem duplication events of triterpene biosynthetic genes function for biosynthesis of oleanane-type ginsenoside and aralosides. They also synthesized 13 aralosides in vivo using *Saccharomyces* system. Overall, the manuscript is well written and show the very unique biosynthesis pathway based on the finding of causal genes using newly assembled high-quality Ae genome sequence. I have a few issues to clarify for this manuscript because the paper can be a golden standard for the genomics research in the Araliaceae family which by far has no clear reference genome sequence.

Response: We thank the reviewer for the careful and overall positive evaluation.

Major Comment 1: The Ae plant might have high heterozygosity. The overall genome assembly might be correct. However, the pseudochromosome sequence is not supported by genetic map or high-resolution FISH analysis using chromosome specific oligomers. Super-scaffolding by Hi-C analysis can induce large scale genome mis-assembly. If some efforts are taken to evaluate the pseudochromosome sequence, the Ae genome sequence can be widely utilized as the reference for comparative study with other Araliaceae genomes.

Response 1: As suggested, we assessed the pseudochromosome of *A. elata* using two methods. (1) We used LAI (LTR Assembly Index) for complete assembly evaluation by using LTR digest, LTR finder and LTR retriever (Ou et al., *Nucleic Acids Research*, (2018) 46 (21): e126). The LAI of the final genome assembly was 19.05, nearly reaching the gold

standard for high-quality genome sequencing and assembly in plants. (2) We demonstrated the quality of the chromosome-level assemblies using the good genome synteny for *E. senticosus* and *P. notoginseng* (**Supplementary Figure 10 and 11**). These results suggest that the assembled genome for *A. elata* does not have any large-scale mis-assembly. We have included these results and descriptions in the revised manuscript (please see page 6, lines 131-134).

For easy reference, the analytical methods are mentioned here.

LAI analysis

First, we found the LTR sequence in the *Aralia elata* genome using LTRdigest (options -minlenltr 100 -maxlenltr 7000 -mintsd 4 -maxtsd 6 -motif TGCA -motifmis 1 -similar 85 -vic 10 -seed 20) and LTR_finder (options -D 15000 -d 1000 -L 7000 -l 100 -p 20 -C -M 0.85). Next, we integrated the results from these two programs to calculate the LAI value using LTR_retriever.

Reference:

Ou, S., Chen, J., Jiang, N. Assessing genome assembly quality using the LTR Assembly Index (LAI), *Nucleic Acids Research* 46, e126 (2018). <https://doi.org/10.1093/nar/gky730>

Major Comment 2: I have concern about Dc-beta genome duplication which was represented in Apiaceae unique genome triplication [or sometime as duplication in the *Eleutherococcus senticosus* genome paper (Yang et al. 2021. *Molecular Ecology Resources*)]. The *Daucus carota* genome paper reported two genome duplications, Dc-beta (triplication) and Dc-alpha (duplication) (Iorizzo et al. 2016. *Nature Genetics*). However, when the *Panax ginseng* (Pg) genome assembly were compared to the *D. carota* genome, they found only two syntenic blocks against four Pg genomes which is derived from Pg-alpha and Pg-beta genome duplications (Kim et. al, 2018 *Plant Biotechnology*). The Ae genome assembly showed clear synteny for Pg-beta with *Vitis* genome which was diverged 120 million years ago. It is clear that Pg-beta genome duplication share with the Ae genome duplication. However, I wonder whether Dc-beta is really an Apiaceae unique event. In this manuscript, there are many genome level comparisons but lack genome comparison between Ae and Dc genomes. I'd like to ask for genome level comparisons to show the Dc-beta genome triplication event to clearly support figures 1 and 2.

Response 2: Thank you very much for your comments. We have also noticed the controversy about the genome duplication events among Apiales genomes. We further analyzed the genome synteny and *Ks* values of paralogs and orthologs among *A. elata*, *D. carota*, and *V. vinifera* genomes. Our studies supported that the Dc-alpha and Dc-beta events are species-specific in *D. carota*, and the *A. elata* WGD event after γ event is shared by Araliaceae species after the split with *D. carota*.

First, the intra-genomic collinearity showed that the *A. elata* experienced only one WGD event (Araliaceae-specific WGD) after the core-eudicot whole-genome triplication (WGT, or γ event), (Supplementary Figure 12). In contrast, the *D. carota* underwent two rounds of polyploidization (Dc-beta and Dc-alpha genome duplications) (Supplementary Figure 13). The median *Ks* value of paralog gene pairs on synteny blocks from *A. elata*

genome was about 0.3, while the Dc-beta and Dc-alpha corresponding to the median *Ks* values of ~1.0 and ~0.6, indicating these WGD events may not occurred close in time (Supplementary Figures 12 and 13).

Second, the well-characterized grape (*V. vinifera*) genome, which is relatively stable genome and is likely not affected by any polyploidization event after the γ event, was used as a reference for inter-genomic dotplot comparison with *A. elata* and *D. carota*. The ratios of the best-matched orthologous regions between two species (*A. elata* and *D. carota*) and *V. vinifera* were 2:1 and 4:1, respectively (Supplementary Figures 14 and 15). This indicates that after divergence from grape, the *D. carota* experienced two additional whole genome duplication events, resulting in overlapping carrot homoeologous regions often up to 4 \times depth. If there had been an extra hexaploidization and tetraploidization event in carrot, as Iorizzo et al. reported (Iorizzo et al. 2016. Nature Genetics), assuming no DNA loss, we would expect a grape gene (or chromosomal region) to have six best-matched or orthologous carrot genes (chromosomal regions). Here, our findings indicated that the Dc-beta event was a tetraploidization instead of a triplication, which is consist with recent reports in Apiaceae (Wang et al., BMC Plant Biology (2020) 20:52; Song et al., Plant Biotechnology Journal (2020) 18:1444–1456; Song et al., Plant Biotechnology Journal (2021) 19(4):731-744).

Hence, assuming the Dc-beta event is shared with *Aralia elata*, we would expect that *A. elata* experienced two additional whole genome duplication events after the γ event and the ratio of the best-matched orthologous regions between *A. elata* and *D. carota* should be 2:2. However, we found a clear 2:4 collinearity relationship between *A. elata* and *D. carota* (Supplementary Figures 16). In summary, our results support that the Dc-beta WGD event (tetraploidization) occurred after the divergence of the Apiaceae and Araliaceae and that the independent WGD event shared in Araliaceae occurred after the Dc-beta WGD. Our results are consistent with the inference on the *E. senticosus* genome (Yang et al. 2021. Molecular Ecology Resources). According to these results, we have modified the Figure 2A and Supplementary Figure 12 and made corresponding modifications in the main text on line 175-179.

Some minor specific comments are listed below:

Minor Comment 1: Line 211: Supplementary Fig 15 \diamond Supplementary Fig. 16

Response 1: Corrected.

Minor Comment 2: Line 217: “post-modification” to “post-transcriptional modification” or “post-translational modification”?

Response 2: We changed “post-modification enzymes” to “tailoring enzymes” on line 233.

Minor Comment 3: Line 218: spelling of “A. ellata”

Response 3: Corrected.

Minor Comment 4: Line 232: wrong figure citation. Fig. 4b \diamond 4c

Response 4: Corrected, line 248.

Minor Comment 15: Line 244 vs 250: CYP72A and CYP72, are they the same? Please be consistent with italics for genes, and please clarify the difference between suffixes like –A.
Response 5: We have corrected these errors, line 260 vs 267.

Minor Comment 6: Fig 3a: Emphasize the two genes (AE11G00256.1 and AE05G00216.1) by making them bold in the phylogenetic tree for easy access.
Response 6: We thank the reviewer for this comment. We bolded the two genes, AeBAS1 (AE05G00216) and AeBAS2 (AE04G00581) in Figure 3b. The gene id of AeBAS2 is AE04G00581, not AE11G00256, and has now been corrected throughout the revised manuscript.

Minor Comment 7: Fig. 4b: You made some CYP716A genes bold to emphasize but the rest (AE11G00717.1 and AE09G02158.1) were not. Any special reason? Please emphasize all for easy visual access.
Response 7: We thank the reviewer for this comment. We have bolded all the CYP716A and CYP72A subfamily genes of *A. elata* in Figure 4b.

Minor Comment 8: Fig. 4c: The yeast strains with the AeBAS1+AeCYP716A1+AeCYP716A2+AeCYP72A2 showed 16-OH-hederagenin peak in the chromatogram, which contradicts the pathway in 4a and your claim in lines 250-252. Based on Fig. 4a, the AeBAS1+AeCYP716A1+AeCYP716A2+AeCYP72A2 construct should only accumulate Echinocystic acid and no 16-OH-hederagenin as CYP72 is unable to process upstream compounds. Is your diagram in Fig. 4a wrong (the names of enzymes, eg. AeCYP72A1/AeCYP72A1)?
Response 8: We thank the reviewer for spotting this error. We have corrected the erroneous gene names in Figure 4a of the revised manuscript.

Minor Comment 9: Fig. 5a,c,e,g: There is no description of what the blue bar means in the legend.
Response 9: We have now described the blue bar in the legend for Fig. 5.

Minor Comment 10: Lines 268-270: Wrong figure reference. There is no genome evolution figure in Fig.5a
Response 10: We thank the reviewer for spotting this error. We have now corrected it on line 286.

Minor Comment 11: Line 280,286: Do you mean Fig. 5c,d?
Response 11: Corrected, lines 299, 305.

Minor Comment 12: Line 292: Wrong figure citation.
Response 12: Corrected, line 311.

Minor Comment 13: Line 295: There is no Fig. 4f. You mean 5f? The use of different

nomenclature is confusing. Please be consistent with what gene names you use. What is AeUGT73CB3? I could not see this in Fig. 5, where you use different gene names, i.e. AE..1544.

Response 13: Corrected, line 315. For consistency, we changed all gene IDs to gene names throughout the manuscript including the figures.

Minor Comment 14: Line 352: Supplementary Fig. 17: Did you mark one gene incorrectly? (AE04G0058.1 is not one of the two selected BAS genes).

Response 14: Sorry, the gene id of *AeBAS2* should be AE04G00581.1 and is now correct throughout the manuscript.

Reviewers' Comments:

Reviewer #1:

Remarks to the Author:

The manuscript has been substantially improved!

I only have one point and that's regarding the UGT assays. The information provided in the rebuttal letter and supplementary data differs from the main manuscript where it is written in line 664 - 667:

The supernatant was used as crude enzyme for enzymatic assays. The 500 μ L enzymatic activity reaction mixture included 50 mM Tris-HCl (pH 7.0), 10 mM MgCl₂, 14 mM 2-mercaptoethanol, 0.1 mM acceptor substrate, 1 mM UDP-Glc and 450 μ L UGT crude protein. The reaction was incubated at 30 °C for 12 h and terminated by adding 500 μ l n-butanol

Reviewer #2:

None

Reviewer #4:

Remarks to the Author:

The authors fully addressed all concerns and added all the additionally required data. I have no further comments.

“Point-by-Point” Responses to Reviewers’ Comments:

Manuscript Number: NCOMMS-21-37280A

Manuscript title: Partial deletion and tandem duplications of biosynthetic genes drive the diversity of triterpene saponins in *Aralia elata*

Reviewer #1 (Remarks to the Author):

Comment 1: The manuscript has been substantially improved!

I only have one point and that's regarding the UGT assays. The information provided in the rebuttal letter and supplementary data differs from the main manuscript where it is written in line 664 - 667:

The supernatant was used as crude enzyme for enzymatic assays. The 500 μ L enzymatic activity reaction mixture included 50 mM Tris-HCl (pH 7.0), 10 mM MgCl₂, 14 mM 2-mercaptoethanol, 0.1 mM acceptor substrate, 1 mM UDP-Glc and 450 μ L UGT crude protein. The reaction was incubated at 30 °C for 12 h and terminated by adding 500 μ L n-butanol.

Response 1: Thanks for pointing this out. The methods used for detecting the biochemical functions (enzymatic activity) and the enzymatic characteristics (enzymatic kinetics) of UGTs were different. Here, we moved the method for enzymatic kinetics from the supplementary data into the main manuscript on lines 677-685.

[Editor: Reviewer #2 states in Remark to Editor section that (s)he is satisfied with the revision.]

[Editor: Reviewer #3 is unavailable. We asked Reviewer #4 to comment on your responses.]

Reviewer #4 (Remarks to the Author):

The authors fully addressed all concerns and added all the additionally required data. I have no further comments.

REVIEWERS' COMMENTS

None.

REVIEWERS' COMMENTS: None.

Response: Thanks.